# Sphingosine-1-phosphate signaling regulates the ability of Müller glia to become neurogenic, proliferating progenitor-like cells

Olivia B Taylor[1,2], Nicholas DeGroff[1], Heithem M El-Hodiri[1], Chengyu Gao[3], Andy J Fischer[1]*

[1]Department of Neuroscience, College of Medicine, The Ohio State University, Columbus, United States; [2]Neuroscience Graduate Program, The Ohio State University, Columbus, United States; [3]Campus Chemical Instrument Center, Mass Spectrometry and Proteomics Facility, The Ohio State University, Columbus, United States

*For correspondence:
Andrew.Fischer@osumc.edu

## eLife Assessment

This **important** study investigates the signaling pathways regulating retinal regeneration. **Convincing** evidence shows that the sphingosine-1-phosphate (S1P) signaling pathway is inhibited following retinal injury. Small-molecule activators and inhibitors support a model in which S1P signaling must be inhibited to generate Müller glial progenitor cells-a key step in retinal regeneration. The presented results support the major conclusions. However, whether the drug treatments directly or indirectly affect the Müller cells remains unclear.

**Abstract** The purpose of these studies is to investigate how Sphingosine-1-phosphate (S1P) signaling regulates glial phenotype, dedifferentiation of Müller glia (MG), reprogramming into proliferating MG-derived progenitor cells (MGPCs), and neuronal differentiation of the progeny of MGPCs in the chick retina. We found that S1P-related genes are highly expressed by retinal neurons and glia, and levels of expression were dynamically regulated following retinal damage. Drug treatments that activate S1P receptor 1 (S1PR1) or increase levels of S1P suppressed the formation of MGPCs. Conversely, treatments that inhibit S1PR1 or decrease levels of S1P stimulated the formation of MGPCs. Inhibition of S1P receptors or S1P synthesis significantly enhanced the neuronal differentiation of the progeny of MGPCs. We report that S1P-related gene expression in MG is modulated by microglia and inhibition of S1P receptors or S1P synthesis partially rescues the loss of MGPC formation in damaged retinas missing microglia. Finally, we show that TGFβ/Smad3 signaling in the resting retina maintains S1PR1 expression in MG. We conclude that the S1P signaling is dynamically regulated in MG and MGPCs in the chick retina, and activation of S1P signaling depends, in part, on signals produced by reactive microglia.

## Introduction

Different cell signaling pathways are known to regulate the reprogramming of Müller glia (MG) into MG-derived progenitor cells (MGPCs) in damaged retinas. There are crucial roles of growth factors, pro-inflammatory cytokines, and transcription factors in the regulation of MG reprogramming into

**eLife digest** The retina is a layered structure of the central nervous system located at the back of our eyes that contains neuronal cells. Retinal neurons convert visible light into nerve signals, which travel from our eyes to our brain along the optic nerve, allowing us to see. Healthy retinas are, therefore, critical for vision.

Unfortunately, many factors can damage our retinas. These range from acute eye injuries to eye diseases like diabetic retinopathy and glaucoma; if too many neurons are damaged, it can lead to blindness. In some animals, damaged retinas can repair themselves, or 'regenerate'. This ability varies depending on the species: in fish, retinal regeneration is highly efficient, but it is reduced in birds and entirely absent in mammals – including humans.

In animals that can regenerate their retinas, the resident support cells of the retina (called Müller glia) respond to retinal injury by dividing to form progenitor cells. These progenitor cells can further reprogram into new neurons to replace damaged tissue and restore sight. The efficiency of this regeneration process depends on how many cells proliferate and the ability of these progenitors to become neurons. In birds, for example, many progenitor cells are formed, but only a fraction turn into neurons.

In the Müller glia of birds – specifically, in chicks – the activity of the genes for a set of biological signals (collectively termed the S1P signalling pathway, or S1P) changes after retinal injury. More generally, S1P is also known to be associated with tissue damage and inflammation. Taylor et al. therefore wanted to determine if S1P played a role in chick retinal regeneration, particularly in the development of Müller glia into 'successful' progenitor cells capable of replacing damaged neurons.

Taylor et al. employed a combination of microscopy and genetic techniques to track the production of various cell types and measure S1P gene activity under different conditions. When chick retinas were treated with drugs known to suppress S1P activity, the production of progenitor cells increased. Importantly, these new progenitor cells were also more likely to develop into new neurons. In contrast, drugs that turned on S1P activity resulted in fewer progenitor cells being formed. These findings suggest that S1P signalling activity suppresses retinal regeneration.

This study adds to our understanding of how inflammatory signals control the retina's ability to repair itself after damage. However, more research is needed to determine the role of S1P in mammalian retinas. Ultimately, Taylor et al. hope that the knowledge gained will produce treatments to recover vision for people with retinal disease.

---

MGPCs in different vertebrate species (*Taylor et al., 2024*). Nuclear Factor Kappa B (NFκB) is one the key cell signaling 'hubs' that regulates the reactivity of MG, acts restore glial quiescence and suppress the neurogenic of MG-derived progenitors in the mouse retina (*Hoang et al., 2020*; *Palazzo et al., 2022*). We found that NFκB signaling in damaged retinas is manifested in MG by pro-inflammatory signals produced by activated microglia (*Palazzo et al., 2022*; *Palazzo et al., 2023*). Further, we show that inhibition of NFκB-signaling results in diminished recruitment of immune cells in damaged retinas, increased neuronal survival, and increased formation of neuron-like cells from Ascl1-overexpressing MG (*Palazzo et al., 2022*). In the chick retina, by comparison, rapid activation of NFκB via signals produced by reactive microglia starts the process of MG activation and reprogramming, but sustained NFκB signaling suppresses the proliferating of MGPCs and neuronal differentiation of progeny (*Palazzo et al., 2020*). NFκB signaling is known to be coordinated with Sphingosine-1-phosphate (S1P) signaling to regulate different cellular processes, including inflammation (*Blom et al., 2010*; *Pérez-Jeldres et al., 2021*; *Zheng et al., 2019*).

S1P signaling is a promising candidate to regulate the ability of MG to reprogram into proliferating, neurogenic progenitor cells. S1P is produced by sphingosine kinases (SPHK1 and SPHK2) and is degraded by a lyase (SGPL1). Notably, SPHK2 does not directly contribute to levels of secreted S1P (*Thuy et al., 2022*), nor is it annotated in the chick genome. S1P can be exported from cells by a transporter (MFSD2A and SPNS2) or converted to sphingosine by a phosphatase (SGPP1). Levels of sphingosine are increased by ASAH1 by conversion of ceramide or decreased by CERS2/5/6 by conversion to ceramide. S1P is known to activate G-protein-coupled receptors, S1PR1 through S1PR5. S1PRs are known to activate different cell signaling pathways including MAPK and PI3K/mTor, and crosstalk with pro-inflammatory pathways such as NFκB (*Hu et al., 2020*). Alternatively, activation of S1PRs can

activate Jak/Stat signaling, which upregulates Egr1 to upregulate pro-inflammatory cytokines that secondarily activate NFκB (*Gurgui et al., 2010*). S1P signaling is an important pathway that has been widely implicated as mediating inflammatory responses, cellular proliferation, promote survival, and regulate angiogenesis (*Obinata and Hla, 2019*). S1P signaling is required for vascular maturation, progenitor cell-cycle exit, and axon guidance in the developing retina (*Simón et al., 2019*). However, S1P can have detrimental effects including migration of MG, angiogenesis, and inflammation associated with proliferative retinopathies and aging. Most of these studies in retinal cells were conducted in vitro, and these studies need to be followed-up by in vivo analyses to assess the coordinated impact of S1P signaling on the many different cell types in the retina. The functions of S1P signaling in normal and damaged retinas are poorly understood, and nothing is known about how S1P signaling impacts MG-mediated retinal regeneration.

Pro-inflammatory signaling represses reprogramming of mouse MG by promoting reactivity networks and anti-neurogenic factors, while increasing cell death in damage retinas (*Palazzo et al., 2022*). Therefore, it is important to understand how different cell signaling pathways impact inflammation to develop techniques to enhance neuronal survival and retinal regeneration. In this study, we target S1P synthesis, S1P degradation and S1P receptors to investigate the roles of S1P signaling during damage-dependent MG reprogramming and neurogenesis in the chick model system.

## Results

### S1P signaling regulates the formation of MGPCs in damaged retinas

We first probed for patterns of expression of S1P-related genes in a large aggregate scRNA-seq library of >180,000 cells isolated from retinas treated with saline, NMDA, or two or three doses insulin + FGF2 and the combination of NMDA and insulin + FGF2 (*Figure 1a*), as described previously (*Clark et al., 2019*; *Campbell et al., 2022*; *El-Hodiri et al., 2022*). MGPC formation can be induced in retinas damaged by NMDA or undamaged retinas treated with repeated doses of insulin + FGF2 (*Fischer, 2005*; *Fischer et al., 2002*). Merging of libraries revealed patterns of gene expression in MG and MGPCs that are treatment-dependent. Retinal cell types ordered into different uniform manifold approximation and projection (UMAP) clusters were identified based on well-established markers (*Figure 1b, c*), as described in the methods. *S1PR1* was most prominently expressed by resting MG and MG returning to a resting state, whereas *S1PR3* was detected in relatively few scattered cells in clusters of MG, ganglion cells, horizontal cells, bipolar cells, amacrine cells, photoreceptors, oligodendrocytes, microglia, and NIRG cells (*Figure 1d*). NIRG cells, or non-astrocytic inner retinal glial cells, are a distinct type of glial cell that arise from optic nerve progenitors (*Rompani and Cepko, 2008*) and have described in retinas of chicks (*Zelinka et al., 2012*) and some species of reptiles (*Todd et al., 2019*). *S1PR2* was not widely expressed in retinal cells (*Figure 1d*). *SPHK1* was detected in scattered cells in all cell types (*Figure 1e*). *SGPL1* was detected in scattered cells in all cell types with prominent expression in rod photoreceptors (*Figure 1e*). *ASAH1* (encoding acid ceramidase enzyme) was prominently expressed in microglia and cone photoreceptors, with expression in cells scattered in all other types of retinal cells (*Figure 1f*). Genes encoding ceramide synthase enzymes showed variable patterns of expression. *CERS6* was prominently detected in all types of inner retinal neurons (*Figure 1f*), whereas *CERS5* was detected in relatively few cells scattered across clusters of different types of cells (*Figure 1f*).

We bioinformatically isolated the MG (>70,000 cells) and re-embedded these cells into a UMAP plot (*Figure 1g*). The UMAP ordering of MG revealed clusters that were comprised of cells from distinct treatments and different times after treatment (*Figure 1h, i*). We find that S1P-related genes are dramatically up- or downregulated in MG in damaged retinas and during the formation of MGPCs. Dynamic changes of mRNA levels are strongly correlated with changes in protein levels and function (*Liu et al., 2016*). We found that levels of *S1PR1* and *CERS6* were very high in resting MG and significantly downregulated in MG and MGPCs from treated retinas (*Figure 1j*; *Figure 1—figure supplement 1d, l*; *Figure 1—source data 1*). *S1PR3* was significantly higher in MGPCs, namely the MGPC2 cluster, compared to MG in all other UMAP clusters (*Figure 1j*; *Figure 1—figure supplement 1f*; *Figure 1—source data 1*). Levels of *SPHK1* were relatively high in resting MG and significantly downregulated in MG and MGPCs from retinas treated with NMDA and/or insulin + FGF2 (*Figure 1j*; *Figure 1—figure supplement 1g*; *Figure 1—source data 1*). *CERS5* was significantly upregulated in

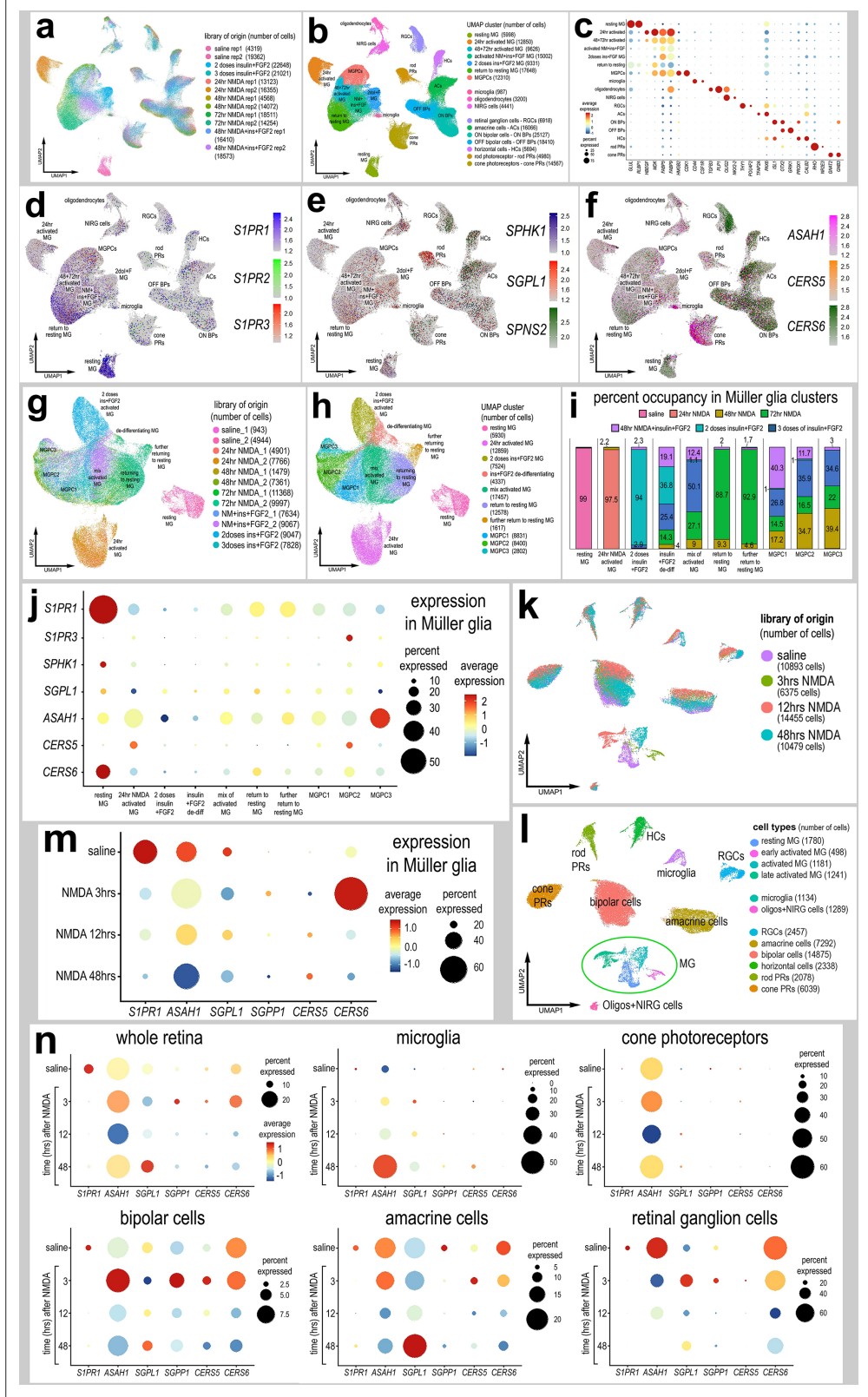

**Figure 1.** Patterns of expression of S1P-related genes. scRNA-seq was used to identify patterns of expression of S1P-related factors among retinal cells with the data presented in UMAP (**a, b, d–h, k, l**) or dot plots (**c, j, m, n**). Aggregate scRNA-seq libraries were generated for cells from (**i**) control retinas and retinas 24, 48, and 72 hr after NMDA-treatment, retinas treated with two or three doses of insulin and FGF2, and retinas treated

*Figure 1 continued on next page*

*Figure 1 continued*

with insulin, FGF2 and NMDA (**a–f**). MG were bioinformatically isolated and analyzed from the large aggregate library (**g–i**), and control retinas and retinas 3, 12, and 48 hr after NMDA (**k–l**). UMAP-ordered cells formed distinct clusters of neuronal cells, resting MG, early activated MG, activated MG, and Müller glia-derived progenitor cells (MGPCs) based on distinct patterns of gene expression (see methods and Figure S1). UMAP heatmap plots illustrate patterns and levels of expression of S1P receptors *S1PR1* and *S1PR3* (**d**), S1P metabolism and transport genes *SPHK1, SGPL1,* and *SPNS2* (**e**), and ceramide metabolism genes *ASAH1, CERS5,* and *CERS6* (**f**) illustrate levels and patterns of expression across the whole retina. Dot plots illustrate relative levels of expression (heatmap) and percent expression (dot size) in MG, activated MG, and MGPCs (**j, m**), whole retina, and other cell types (**n**) in different UMAP clusters. Significance of difference was determined by using a Wilcox rank sum with Bonferoni correction. Abbreviations: MG – Müller glia, NMDA – *N*-methyl-D-aspartate, UMAP – uniform manifold approximation and projection.

The online version of this article includes the following source data and figure supplement(s) for figure 1:

**Source data 1.** Statistics for S1P-related genes in aggregate scRNA-seq libraries.

**Figure supplement 1.** Patterns of expression of genes in resting MG, activated MG, and Müller glia-derived progenitor cells (MGPCs).

---

MG at 24 hr after NMDA and MGPCs compared to resting MG and MG treated with insulin and FGF2 (*Figure 1j*; *Figure 1—figure supplement 1k*; *Figure 1—source data 1*).

We next analyzed the expression of S1P-related genes in a scRNA-seq database that we generated for early time-points, 3 and 12 hr, after NMDA-treatment (*Clark et al., 2019*; *Campbell et al., 2022*; *El-Hodiri et al., 2023*). These scRNA-seq libraries did not integrate well with older libraries likely because these libraries were generated with different reagents with different sensitivities. UMAP ordering of cells revealed distinct clusters of neurons and glia, with MG forming distinct clusters based on time after NMDA-treatment (*Figure 1k, l*). Resting MG were identified based on high levels of expression of genes such as *GLUL* and *RLBP1*, activated MG were identified based of high levels of expression of genes such as *TGFB2* and *PMP2*, and proliferating MGPCs were identified based on high levels of expression of *CDK1* and *TOP2A* (*Figure 1—figure supplement 1m–q*). Consistent with previous observations, *S1PR1* was highly and exclusively expressed by resting MG and is rapidly (<3 hr) downregulated following NMDA-induced damage (*Figure 1m*; *Figure 1—figure supplement 1r*). Levels of *SPHK1* were very low in this aggregate scRNA-seq library possibly because the transcripts are very low copy or the mRNA is labile (not shown). *SGPL1* and *ASAH1* are rapidly downregulated, and *CERS6* is upregulated at different times after NMDA-treatment (*Figure 1e, f, m, n*), suggesting a downregulation of S1P levels following damage in the chick retina.

In microglia, levels of *ASAH1* and *SGPL1* were significantly increased at 48 hr after NMDA-treatment (*Figure 1n*; *Figure 1—source data 1*). In cone photoreceptors, only ASAH1 was significantly decreased following NMDA-treatment (*Figure 1*; *Figure 1—source data 1*). In bipolar cells, there were significant decreases in levels of *S1PR1* and *CERS6* following NMDA-treatment (*Figure 1n*; *Figure 1—source data 1*). In amacrine cells, there were significant decreases in levels of *S1PR1*, *ASAH1*, and *CERS6* following NMDA-treatment (*Figure 1n*; *Figure 1—source data 1*). In RGCs, there were significant decreases in expression levels of CERS6 and SGPL1 at different times after NMDA-treatment (*Figure 1*; *Figure 1—source data 1*). There were no significant changes in S1P-related genes in rod photoreceptors or horizontal cells in damaged retinas (not shown). Collectively, these findings indicate that there is dynamic expression of genes related to S1P synthesis and degradation in retinal neurons, and this may impact signaling through *S1PR1* in MG.

## Validation of patterns of expression of *S1PR1*, *S1PR3*, and *SPHK1*

To validate some of the findings from scRNA-seq libraries, we performed fluorescence in situ hybridization (FISH) on normal and NMDA-damaged retinas. We applied antibodies to S1pr1 and Sphk1 (Santa Cruz, sc-48356; Novus, NB120-11424; Bioss, bs-2652R), but these antibodies did not reveal plausible patterns of labeling (not shown). By comparison, FISH for *S1PR1* revealed distinct puncta that were concentrated around the Sox2-positive nuclei of resting MG in undamaged retinas, consistent with scRNA-seq data (*Figure 2a, b*). 100% of the Sox2-positive MG nuclei in the inner nuclear layer (INL) were associated with *S1PR1* FISH puncta (*Figure 2b*). In damaged retinas at 3 HPI (hour post injury), *S1PR1* was significantly reduced in the INL and very few puncta were observed in close

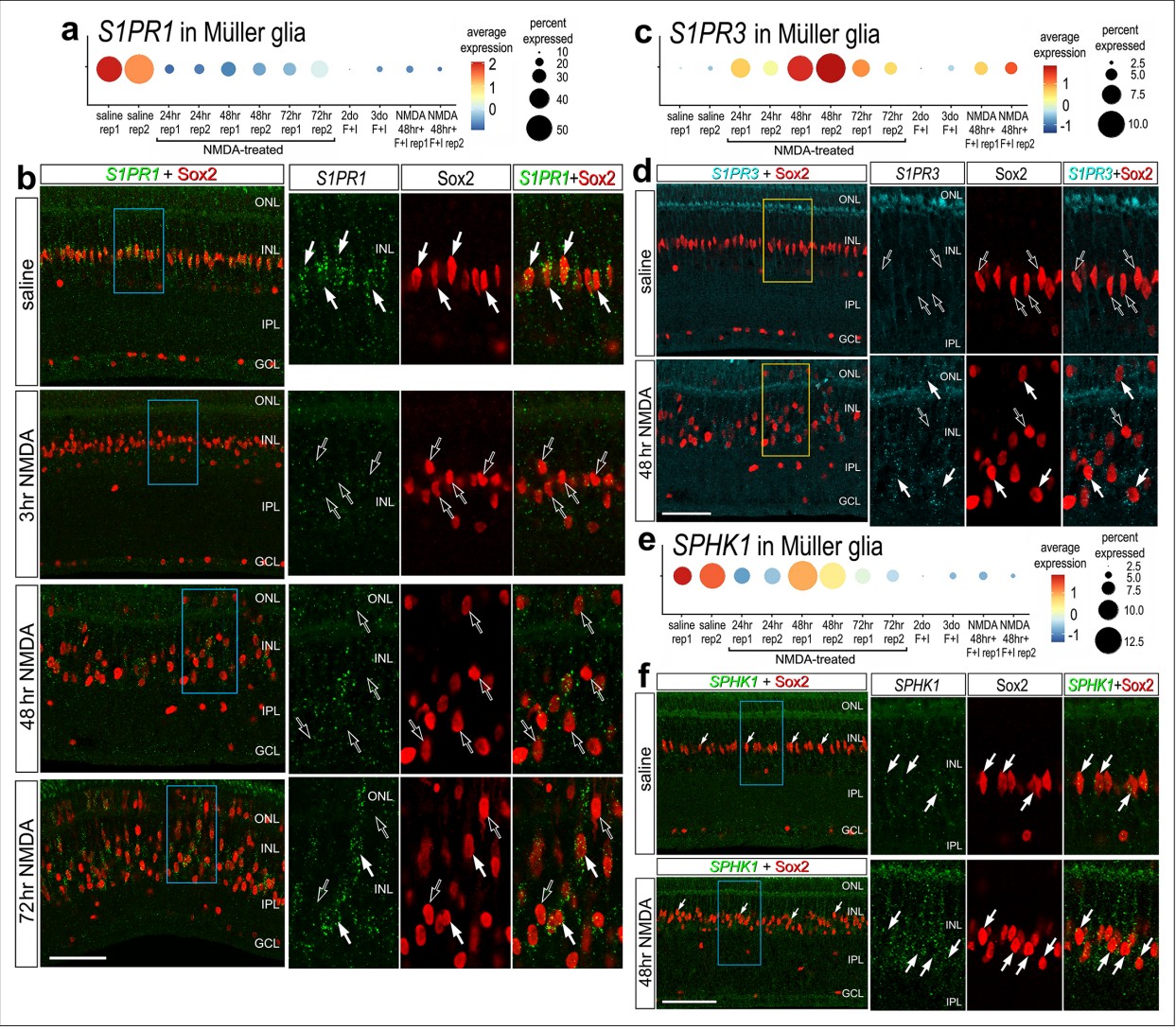

**Figure 2.** Fluorescence in situ hybridization (FISH) for *S1PR1*, *S1PR3*, and *SPHK1*. Retinas were obtained from undamaged, saline-treated eyes and eyes injected with NMDA at 3, 48, or 72 hr after treatment. Retinal sections were labeled with antibodies to Sox2 (red) or FISH probes to *S1PR1* (green puncta; **b**), *S1PR3* (cyan puncta; **d**), or *SPHK1* (green puncta; **f**). Dot plots illustrate the average expression (heatmap) and percent expressed (dot size) for *S1PR1*, *S1PR3*, and *SPHK1* in Müller glia (MG) from different treatments and replicates (rep1 or rep2) (**a, c, e**). Significance of difference was determined by using a Wilcox rank sum with Bonferoni correction. Hollow arrows indicate MG nuclei labeled for Sox2 alone and solid arrows indicate MG nuclei associated with numerous FISH puncta. Calibration bars in panels **b**, **d**, and **f** represent 50 μm. Areas indicated by cyan or yellow are enlarged twofold in adjacent panels. Abbreviations: ONL – outer nuclear layer, INL – inner nuclear layer, IPL – inner plexiform layer, GCL – ganglion cell layer, ns – not significant, NMDA – *N*-methyl-D-aspartate.

proximity to MG nuclei (*Figure 2b*). At 48 and 72 HPI, *S1PR1* signal increased and was observed with Sox2+ nuclei of MG that were delaminated away from the middle of the INL (*Figure 2b*). In undamaged retinas, we found very few *S1PR3* FISH puncta associated with MG nuclei and across the retina (*Figure 2c, d*). In contrast, we identified *S1PR3* puncta associated with some MG nuclei at 48 HPI; these results are consistent with scRNA data (*Figure 2d*). In undamaged retinas, we found *SPHK1* FISH puncta associated with MG nuclei, consistent with scRNA-seq data (*Figure 2e, f*). At 48HPI, we observed an increase in *SPHK1* FISH puncta associated with delaminating MG nuclei, which was similar to findings from scRNA-seq data (*Figure 2e, f*).

## Activation of cell signaling pathways by S1P

S1P signaling is known to activate different cell signaling pathways including MAPK and PI3K/mTor, and crosstalk with pro-inflammatory pathways such as NFκB (reviewed by *Cui et al., 2022*). Alternatively,

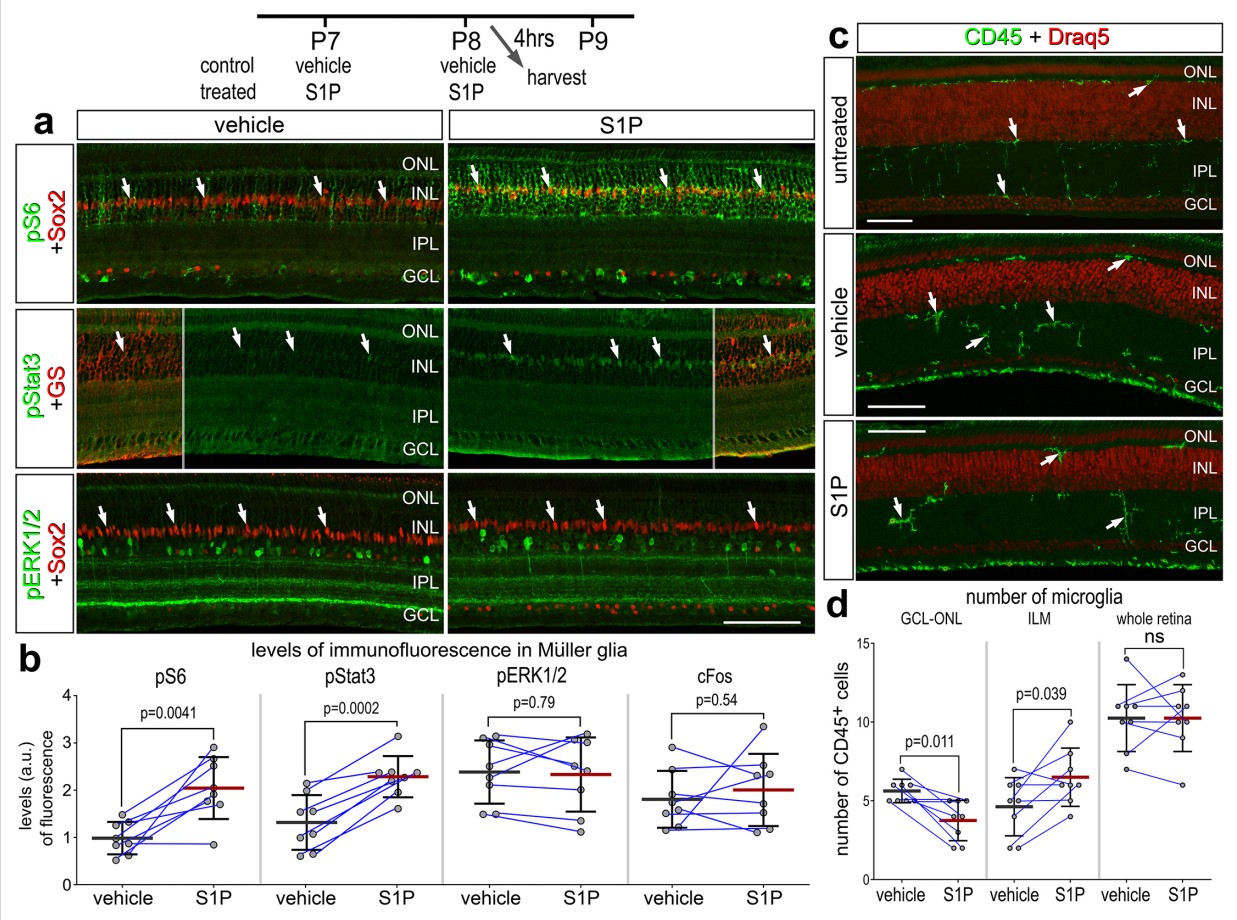

**Figure 3.** Activation of cell signaling pathways and microglia by S1P. Retinas were obtained from eyes injected with vehicle or S1P at P7 and P8, and harvested 4 hr after the last injection. Retinal sections were labeled with antibodies to Sox2 (red) and pS6 (green), pStat3 (green), or pERK1/2 (**a**), or antibodies to CD45 (green) or DNA stain (Draq5; red) (**c**). Arrows indicate the nuclei of Müller glia (MG). Calibration bars in panels **a** and **c** represent 50 μm. Areas indicated by cyan or yellow are enlarged twofold in adjacent panels. The histograms in (**b**) illustrate the mean (bar ± SD) fluorescence intensity sum for pS6, pStat3, pERK1/2, and cFos in MG. The histograms in (**d**) illustrate the mean (bar ± SD) numbers of CD45+ microglia in the GCL–ONL, at the INL, or whole retina. Each dot represents one biological replicate and blue lines connect replicates from control and treated retinas from one individual. Significance of difference (p-values) was determined by using a paired *t*-test. Abbreviations: ONL – outer nuclear layer, INL – inner nuclear layer, IPL – inner plexiform layer, GCL – ganglion cell layer, ns – not significant.

activation of S1PRs can activate Jak/Stat signaling, which upregulates the immediate early gene Egr1 (*Gurgui et al., 2010*; *Liang et al., 2013*; *Sato et al., 2000*). To probe for the activation of different cell signaling pathways in the retina, we delivered S1P to undamaged eyes and collected retinas 24 hr later. In retinas treated with exogenous S1P, levels of pS6 and pSTAT3 levels were significantly increased in MG (*Figure 3a, b*). By comparison, levels of pERK1/2 and cFos were unaffected by S1P-treatment (*Figure 3a, b*). We did not observe significant changes in the total number of microglia in the retina, indicating that S1P delivery did not initiate any recruitment of macrophages into the retina (*Figure 3c, d*). However, we found that S1P-treatment significantly increased the accumulation of microglia to the inner limiting membrane (*Figure 3c, d*). Collectively, these data indicate that the signaling pathways activated by S1P include Jak/Stat and mTOR, but not MAPK. These pathways are predominantly activated in MG, and high levels of S1P may elicit a migratory response in resident retinal microglia. S1P receptor transcripts are not highly expressed in microglia (see *Figure 1*); it is most probable that this response is elicited indirectly through S1PRs expressed by MG and secondary production of chemotactic signaling. There are no reliable cell-level readouts of NFκB signaling in the chick retina (*Palazzo et al., 2020*). Thus, we did not probe for activation of NFκB in retinas treated with S1P.

## S1P signaling and the formation of proliferating MGPCs

We next investigated how activation or inhibition of S1P-signaling influences the formation of proliferating MGPCs in damaged retinas. To test how S1P-signaling influences the formation of MGPCs, we applied different small molecule agonists and antagonists to different enzymes and receptors involved in S1P signaling before, with and after NMDA-treatment (*Figure 4a*). We found significantly reduced numbers of EdU$^+$Sox2$^+$ MGPCs in damaged retinas treated with S1PR1 agonist (SEW2871) (*Figure 4a, c*). Conversely, we found significantly increased numbers of proliferating MGPCs in damaged retinas treated with S1PR1 antagonists (MT1303 or NIBR0213) (*Figure 4a, d, e*). Similarly, S1PR1/3 inhibition (VPC23019) increased numbers of proliferating MGPCs (*Figure 4a, f*), but S1PR3 inhibition alone (TY52156) had no effect on numbers of proliferating MGPCs (data not shown).

We also applied FTY720 (Fingolimod), which has been used in clinical applications. FTY720 is known to act as an agonist at S1P receptors and then induce internalization *and* deactivation after continuous exposure, whereas other agonists are believed to induce persistent changes in signaling through S1P receptors (*Jo et al., 2005*; *Sykes et al., 2014*). FTY720 treatment significantly reduced numbers of proliferating MGPCs (*Figure 4a, h*). Finally, we targeted the metabolism of S1P with inhibitors to SPHK1 or SGPL1, enzymes that synthesize and degrade S1P, respectively. Inhibition of SPHK1 (PF543 or SKI-II), which is expected to decrease levels of S1P, significantly increased numbers of proliferating MGPCs (*Figure 4a, h, i*) By comparison, inhibition of SGPL1 with S1PL-in-31, which is expected to increase levels of S1P, significantly decreased in numbers of proliferating MGPCs (*Figure 4a, b, j*).

To corroborate findings from EdU incorporation studies, we probed for numbers of MGPCs that were labeled for neurofilament, Sox9 and phospho-histone H3 (pHisH3). MGPCs are known to upregulate neurofilament and pHisH3 during late G2 and M phase of the cell cycle at 3 days after NMDA-treatment (*Fischer et al., 2002*; *Zelinka et al., 2016*; *Palazzo et al., 2020*). Consistent with findings from EdU incorporation, we found significant increases in numbers of MGPCs labeled for Sox9, neurofilament and pHisH3 in damaged retinas treated with S1PR1 antagonist (MT1303) or SPHK1 antagonist (PF543) (*Figure 4k, m, o*). Further, we found significant decreases in numbers of MGPCs labeled for Sox9, neurofilament and pHisH3 in damaged retinas treated with S1PR1 modulator (FTY720) or SGPL1 antagonist (S1PL-in-31; *Figure 4n, p*).

None of the agonists or antagonists had a significant effect on the proliferation of microglia/macrophage in NMDA-damaged retinas (*Figure 4—figure supplement 1*). Similarly, except for the S1PR1 antagonist MT1301, none of the agonists or antagonists significantly affected numbers of dying cells (*Figure 4—figure supplement 2*). In damaged retinas treated with MT1303 we found a significant decrease in numbers of TUNEL-positive cells in the inner INL (*Figure 4—figure supplement 2a, c*). In summary, treatments that increase levels of S1P or activate signaling through S1P receptors result in decreased proliferation of MGPCs, whereas treatments that decrease levels of S1P or inhibit signaling through S1P receptors result in increased proliferation of MGPCs, without significant effects upon cell death or the proliferation/accumulation of microglia.

Ceramide synthesis and ceramidase genes have variable patterns of expression in the resting and damaged chick retina (*Figure 1*). S1P and ceramides have been described as a rheostat that regulates cell survival; S1P is protective, and ceramides trigger apoptotic pathways (*Simón et al., 2019*). ASAH1 (acid ceramidase) acts to restrict ceramide accumulation and is dynamically regulated in the chick retina (*Figure 1*). To investigate whether ceramide accumulation regulates cell survival and MGPC proliferation, we applied an acid ceramidase inhibitor (Ceranib-2) before and following NMDA-treatment (*Figure 4—figure supplement 3*). We observed no significant differences in cell death, microglia reactivity, MGPC proliferation between Ceranib-2-treated retinas and control damaged retinas (*Figure 4—figure supplement 3*). Thus, ceramide accumulation may not impact cell survival or the responses of MG to retinal damage.

MG reprogramming remains robust in mature chick retinas from P7 and up to P30, but the zones of proliferation become increasingly confined to the peripheral regions of retina (*Fischer and Reh, 2003b*). In post-hatch chick retina, it is likely that MG 'maturation' occurs in a central-to-peripheral gradient, much like the process neuronal differentiation in the embryonic retina. Accordingly, a zone of regeneration-competent MG persists in the retinal periphery in maturing chicks. We have previously shown that the expression of Notch-related genes differs by retinal region and chick age, and Notch inhibition suppresses numbers of MGPCs in central but not peripheral regions (*Ghai et al., 2010*). By comparison, we observed that *SPHK1* expression was low in the central retina at 48 HPI,

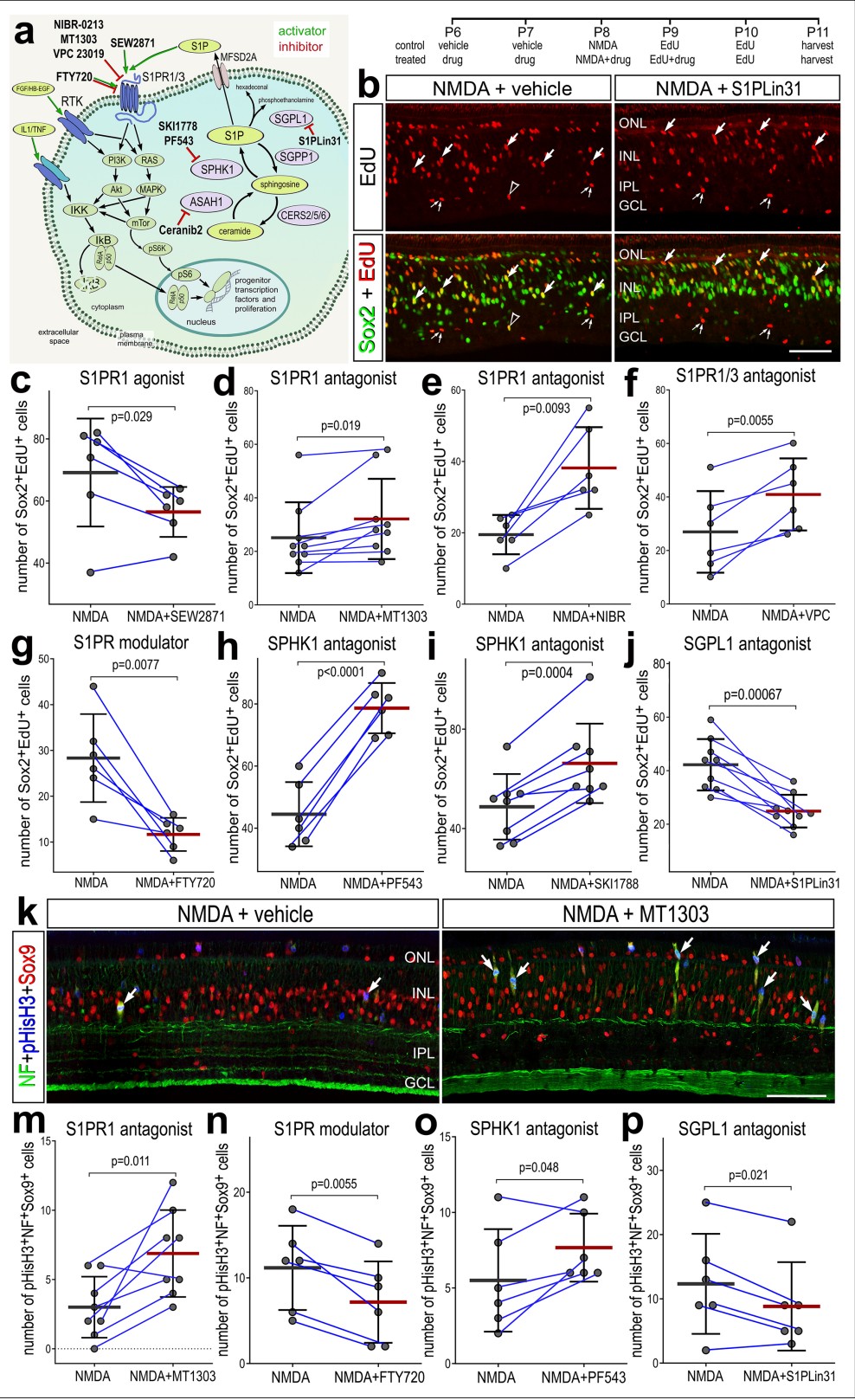

**Figure 4.** Effects of different inhibitors to S1P receptor, synthesizing enzymes and degrading enzymes on the formation of proliferating Müller glia-derived progenitor cells (MGPCs). Schematic diagram of the receptors and enzymes involved in S1P signaling (**a**). The diagram includes the different agonists and antagonists that were applied to target the different components of the S1P pathway (**a**). Eyes were injected with vehicle or drug at

*Figure 4 continued on next page*

*Figure 4 continued*

P6 and P7, NMDA ± drug at P8, EdU ± drug at P9, EdU at P10, and retinas harvested at P11. Retinal sections were labeled for EdU incorporation and antibodies to Sox2 (**b**) or Sox9, neurofilament, and phospho-histone H3 (**k**). Arrows indicate Müller glia (MG), small double arrows indicate presumptive EdU-labeled microglia, and hollow arrow-heads indicate presumptive proliferating NIRG cells. The calibration bar represents 50 μm (**b, k**). Histograms (**c–j, m–p**) illustrate the mean (bar ± SD), each dot represents one biological replicate, and blue lines connect counts from control and treated retinas from one individual. Significance of difference (p-values) was determined by using a paired *t*-test. Abbreviations: ONL – outer nuclear layer, INL – inner nuclear layer, IPL – inner plexiform layer, GCL – ganglion cell layer, ns – not significant, NMDA – *N*-methyl-D-aspartate, EdU – 5-ethynyl-2-deoxyuridine.

The online version of this article includes the following figure supplement(s) for figure 4:

**Figure supplement 1.** Microglia proliferation was unaffected by agonists and antagonists to different components of S1P signaling.

**Figure supplement 2.** Numbers of dying cells are unaffected by agonists and antagonists to different components of S1P signaling.

**Figure supplement 3.** ASAH1 inhibition did not influence proliferating Müller glia-derived progenitor cells (MGPCs), proliferating microglia, or cell death.

**Figure supplement 4.** *SPHK1* expression is suppressed in the central retina.

---

whereas *SPHK1* expression was high in peripheral retina (*Figure 4—figure supplement 4a*). Patterns of *S1PR1* expression in MG were similar in central and peripheral retinal regions (data not shown). Consistent with these patterns of expression, we observed increased numbers of MGPCs in central retinas treated with S1PR1 inhibitor, whereas SPHK1 inhibitor had no effect on numbers of proliferating MGPCs in the central retina (*Figure 4—figure supplement 4b*). We conclude that as MG mature the ability to upregulate *SPHK1* in response to damage is lost in central regions of the retina.

## Cell signaling downstream of SPHK1 inhibition in damaged retinas

There are several known downstream targets of S1PR1 signaling which may be active in MG after retinal damage. To identify second messenger pathways that were affected by diminished levels of S1P in damaged retinas, we applied SPHK1 inhibitor before, with and after NMDA-treatment and collected retinas 24 HPI. We found a significant decrease in levels of ATF3, pS6, and pSmad1/5/9 in MG in damaged retinas treated with SPHK1 inhibitor (*Figure 5a–e*). By comparison, phosphorylated ERK1/2 and cFos were not significantly different in MG in damaged retinas treated with SPHK1 inhibitor (*Figure 5—figure supplement 1*). These findings were relatively consistent with our previous findings, where mTOR activation was observed with exogenous S1P-treatment, but there was no change to pERK1/2 or immediate early gene cFOS (*Figure 5—figure supplement 1*).

Next, we sought to validate S1P levels using liquid chromatography–mass spectrometry (LC/MS). We collected whole retinas 48 hr after saline treatment, Sphk1 inhibitor (PF543) treatment, NMDA-treatment, NMA + PF543 treatment, or NMDA + S1P lyase inhibitor (S1PLin31) treatment. As predicted, Sphk1 inhibitor treatment decreased levels of S1P in undamaged retinas (*Figure 5e*). We found that S1P levels increased in NMDA-treated retinas, which correlated well with increased Sphk1 transcript levels after damage. Surprisingly, SPHK1 inhibitor treatment was not sufficient to ameliorate high levels of S1P induced by NMDA. It is possible that the effects of PF-543 ($k_{off}$ $t_{1/2}$ = 8.5 min) subsided in the 24 hr prior to tissue collection. Alternatively, this observation may reflect the rapid and transient nature of SPHK1 activity after damage, which is consistent with prior experimental observations; we found that S1P/S1PR1-targeting drugs had to be applied prior to NMDA damage to produce robust effects on MGPC proliferation (data not shown). S1P lyase inhibitor treatment robustly boosted levels of S1P in damaged retinas (*Figure 5f*). In summary, differences in levels of S1P between undamaged retinas and NMDA-treatment are consistent with observed Sphk1 RNA levels (*Figure 2e, f*). Further, we validated the targeting activity of two drugs, PF543 and S1PLin31.

## Inhibition of S1P signaling and neurogenesis

We next investigated whether S1P signaling affects the neural differentiation of MGCPs after NMDA-treatment. We find that levels of S1PR1 were rapidly downregulated in activated MG but later rose in MG returning to resting, and S1PR3 was highly expressed in a percent of MGPCs (see *Figure 1*). We

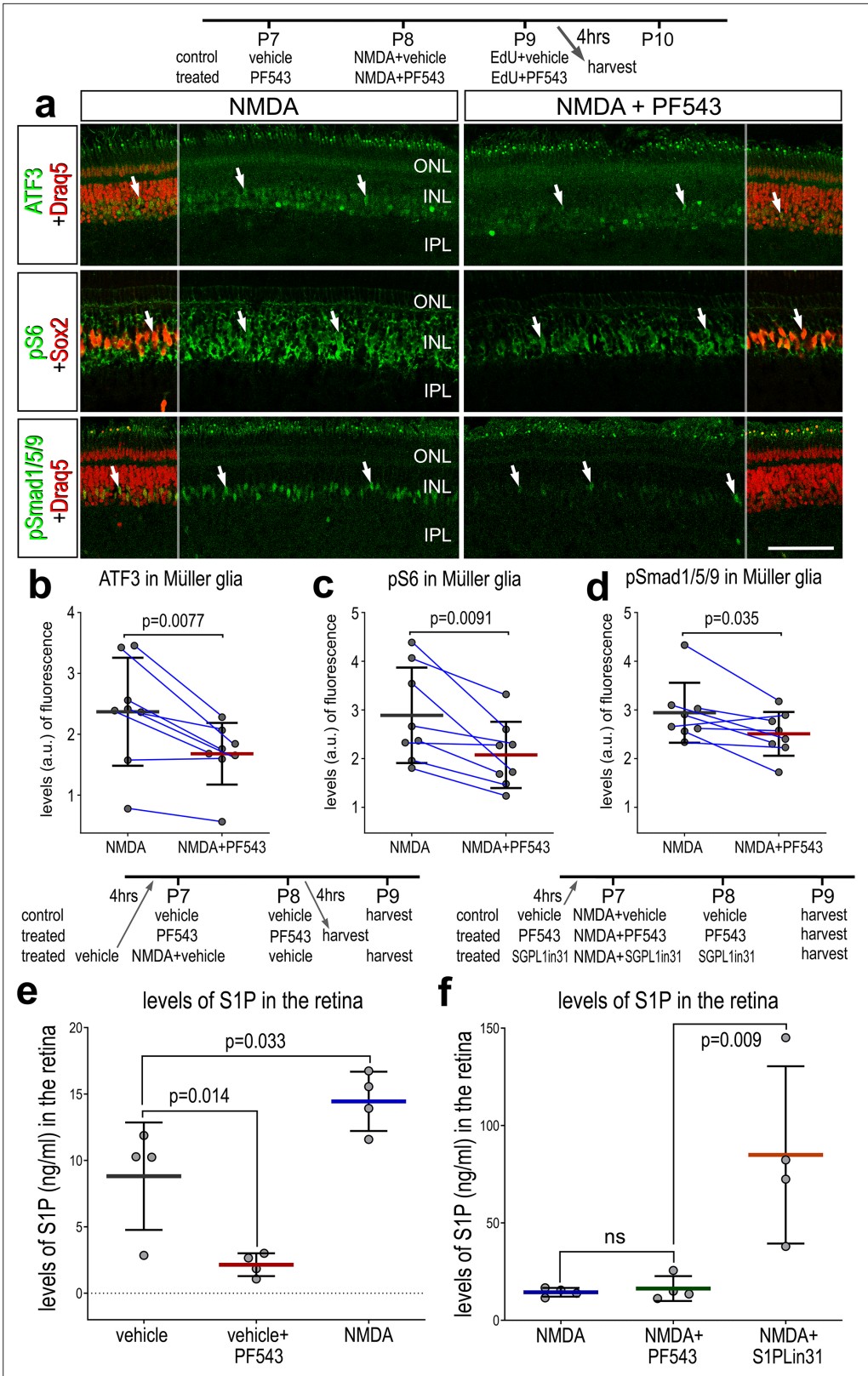

**Figure 5.** Effects of Sphk1 inhibitor on on S1P levels and activation of cell signaling pathways. Eyes were injected with vehicle or PF543 at P7, NMDA ± PF543 at P8, EdU ± PF543 P9, and retinas harvested 4 hr after the last injection. Sections of the retina were labeled for DNA (Draq5; red), ATF3 (green), pS6 (green), Sox2 (red), and pSmad1/5/9 (green) (**a**). Draq5 and Sox2 are included as a partial-field overlay (red, **a**). Arrows indicate the nuclei

*Figure 5 continued on next page*

*Figure 5 continued*
of Müller glia (MG). Calibration bar represents 50 μm (**a**). Eyes were injected with vehicle, PF543, or NMDA at P7, treated with vehicle or PF543 at P8, and harvested 4 hr later (**e**). LC/MS was used to quantify levels of S1P in retinas treated with saline, NMDA, PF543, NMDA + PF543, or NMDA + SGPL-in-31. Histograms (**b–f**) illustrate the mean (bar ± SD), each dot represents one biological replicate and blue lines connect counts from control and treated retinas from one individual. Significance of difference (p-values) in **b**, **c**, and **d** was determined by using a paired *t*-test. Significance of difference (p-values) in **e** and **f** was determined using a one-way ANOVA with Šidák correction for multiple comparisons. Abbreviations: ONL – outer nuclear layer, INL – inner nuclear layer, IPL – inner plexiform layer, GCL – ganglion cell layer, ns – not significant, NMDA – *N*-methyl-D-aspartate, EdU – 5-ethynyl-2-deoxyuridine.

The online version of this article includes the following figure supplement(s) for figure 5:

**Figure supplement 1.** cFos and pERK1/2 levels were unaffected by Sphk1 inhibitor treatment in damaged retinas.

hypothesized that S1P receptor activity during this period of cell-fate specification of MGPC progeny may restrict neural differentiation. We applied S1P pathway inhibitors starting 2 days after NMDA, after MG have committed to cell-cycle re-entry (*Fischer and Reh, 2003a*). We found that SPHK1, S1PR1, and S1PR1/3 inhibitors applied after NMDA had no significant effect on the proliferation of MGPCs (*Figure 6a, b*). We found a significant increase in EdU⁺ amacrine-like cells that express HuC/D (*Figure 6c, d*). There was no significant difference in new Calretinin+ (*Figure 6e, f*) or Ap2a⁺ neurons (not shown) across treatment groups. Further, we found that while many EdU⁺ progeny migrated distally into the ONL, these cells did not co-label for the photoreceptor marker visinin (*Figure 6g*). These findings suggest that inhibiting S1P synthesis or receptor activity enhances the neural differentiation of MGPC progeny to differentiate into HuC/D⁺ amacrine-like neurons after retinal damage. However, the increase in differentiation of amacrine-like cells did not include differentiation of subsets of amacrine cells that express Ap2a or calretinin.

## Inhibition of S1P signaling rescues MGPC proliferation in damaged retinas missing microglia

In chicks and zebrafish, depletion of microglia prior to damage stunts the formation of MGPCs and neural regeneration in the retina (*Fischer et al., 2014*; *Huang et al., 2012*). We have previously identified several factors that can 'rescue' reprogramming in the absence of microglia, including NFκB activators, FGF2, and HBEGF (*El-Hodiri et al., 2023*; *Palazzo et al., 2020*). In addition, we have found increased numbers of dying cells in NMDA-damaged retinas missing microglia (*Todd et al., 2019*). Given that S1P signaling is known to be involved in regulating inflammatory responses involving immune cells (*Obinata and Hla, 2019*), we sought to determine whether the ablation of microglia/macrophages from the retina influenced the expression of S1P-related genes. Accordingly, we probed scRNA-seq libraries of normal and damaged retinas with and without microglia (saline and clodronate saline, respectively); the generation of these libraries has been described previously (*El-Hodiri et al., 2023*), In short, eyes were treated with saline or clodronate liposomes at P6, treated with saline or NMDA at P10, and retinas were harvested at P11. Consistent with previous reports (*Fischer et al., 2014*; *Zelinka et al., 2012*), a single intravitreal injection of clodronate liposomes effectively eliminated >99% of microglia from the retina (not shown). Analyses of scRNA-seq libraries of MG from undamaged and NMDA-damaged retinas with and without microglia showed that many components of S1P signaling were upregulated in resting MG in undamaged retinas (*Figure 7a–l*; *Figure 7—source data 1*). These genes included *S1PR1*, *S1PR3*, *SPHK1*, *SGPL1*, and *ASAH1* (*Figure 7d–i*). Notably, *S1PR1* expression was significantly diminished in undamaged retinas treated with clodronate (microglia-ablated) (*Figure 7d, j*; *Figure 7—source data 1*). Interestingly, *CERS6* was significantly upregulated in MG and other retinal cells in undamaged saline-clodronate retinas (*Figure 7j–m*; *Figure 7—source data 1*). In MG other S1P-related factors that were up- or downregulated with NMDA-treatment were not significantly affected by the ablation of microglia. However, in neuronal cells many factors (including *SPGL1*) normally upregulated with NMDA showed diminished levels of expression when microglia were absent, indicating that microglial reactivity is a driver of S1P metabolism in retinal neurons (*Figure 7k–m*; *Figure 7—source data 1*).

We next tested whether intravitreal injections of inhibitors to SPHK1 and S1PR1 rescued the deficit in MGPC proliferation and increased cell death in damaged retinas missing microglia. As

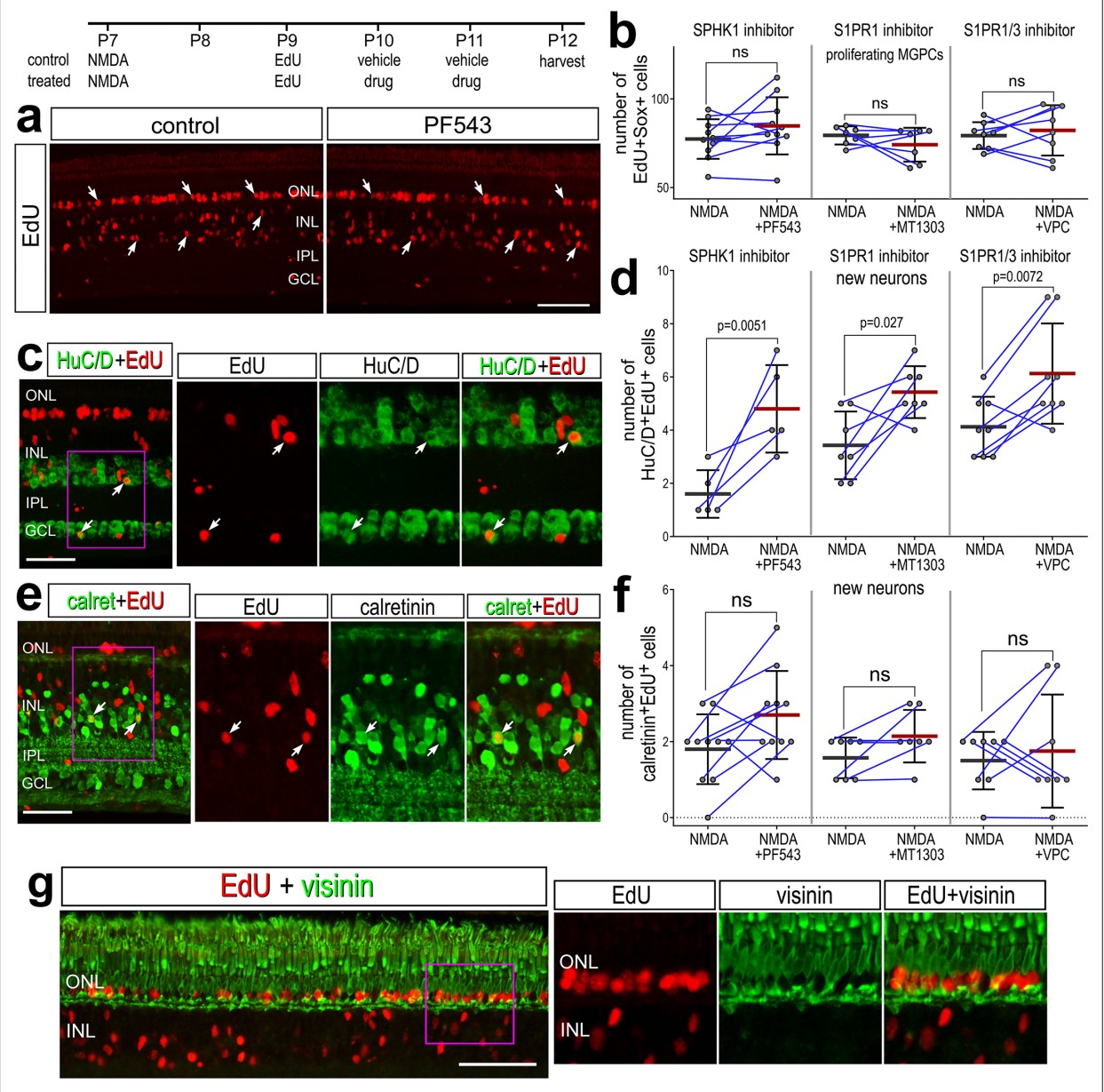

**Figure 6.** Inhibition of S1P synthesis and S1P receptors stimulates neuronal differentiation from Müller glia-derived progenitor cells (MGPCs). Eyes were injected with NMDA at P7, EdU at P9, vehicle or drug at P10 and P11, and retinas harvested at P12. Retinas sections were labeled for EdU (red; **a, c, e, g**) and antibodies to HuC/D (green; **c**), calretinin (green; **e**), or visinin (green; **g**). Arrows indicate the nuclei of regenerated neurons. Calibration bars in panels **a, c, e,** and **g** represent 50 μm. Histograms represent the mean (bar ± SD) and each dot represents one biological replicate (blue lines connect data points from control and treated retinas from the same individual) (**b, d, f**). Significance of difference (p-values) was determined by using a paired *t*-test. Abbreviations: ONL – outer nuclear layer, INL – inner nuclear layer, IPL – inner plexiform layer, GCL – ganglion cell layer, ns – not significant, NMDA – *N*-methyl-D-aspartate, EdU – 5-ethynyl-2-deoxyuridine.

described previously, delivery of clodronate liposomes prior to damage dramatically reduced numbers of EdU⁺ MGPCs (*Figure 8a–d*). However, injections of S1PR1 or SPHK1 inhibitors with and following NMDA significantly increased numbers of proliferating MGPCs in retinas missing microglia (*Figure 8a–d*). However, the number of proliferating MGPCs in S1P inhibitor-treated retinas was significantly lower than in NMDA-treated retinas with activated microglia (*Figure 8b–d*), indicating a partial rescue of MGPC proliferation. Reactive microglia appear to support neuroprotection in the mouse retina (*Todd et al., 2019*), but exacerbate NMDA-induced cell death in the chick retina (*Fischer et al., 2015*). Interestingly, we found that there were differences in cell death between

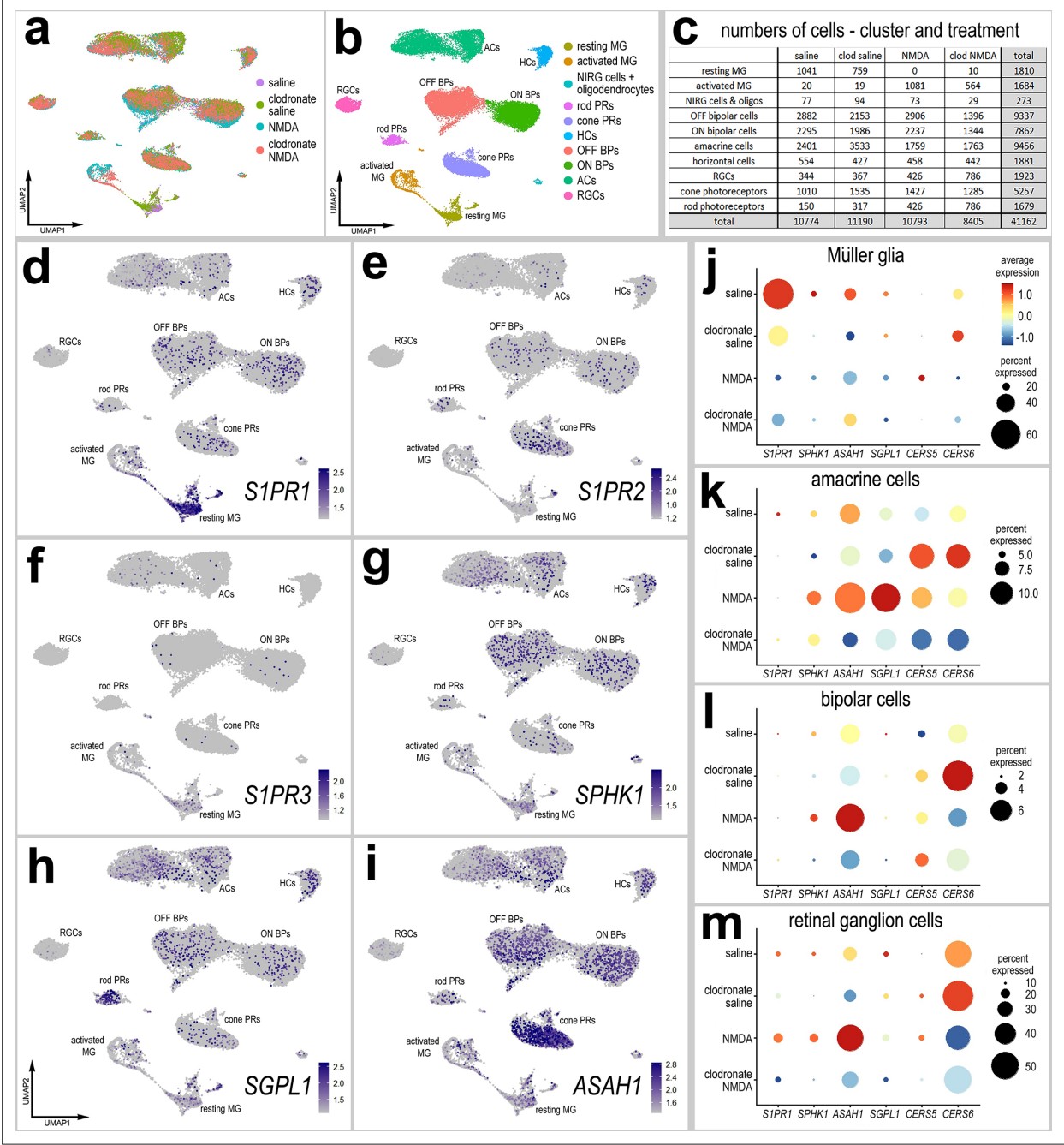

**Figure 7.** scRNA-seq of normal and damaged retinas with and without microglia. Retinas were obtained from eyes injected with saline or clodronate liposomes at P6, saline or NMDA at P10, and tissues harvested at 24 hr after the last injection. UMAP ordering of cells is displayed for libraries of origin and distinct clusters of cells (**a, b**). Number of cells in each UMAP cluster and library of origin are listed in panel **c**. The identities of clusters of cells in UMAP plots were well-established markers (see methods). UMAP heatmap plots illustrate the patterns and levels of expression of *S1PR1*, *S1PR2*, *S1PR3*, *SPHK1*, *SGPL1*, and *ASAH1* (**d–i**). Different cell types, including Müller glia (MG) (**j**), amacrine cells (**k**), bipolar cells (ON and OFF cells; **l**), and retinal ganglion cells (RGCs; **m**) were bioinformatically isolated and dot plots were generated to assess levels of expression by treatment (library of origin). Dot plots illustrate the percentage of expressing cells (dot size) and significant (p < 0.01) changes in expression levels (heatmap) for genes for cells from retinas treated with saline versus saline-clodronate and NMDA versus NMDA-clodronate. Significance of difference was determined by using a Wilcox rank sum with Bonferoni correction. Abbreviations: NMDA – *N*-methyl-D-aspartate, UMAP – uniform manifold approximation and projection.

The online version of this article includes the following source data for figure 7:

**Source data 1.** Statistics for S1P-related genes in clodronate- or saline-treated aggregate scRNA-seq libraries.

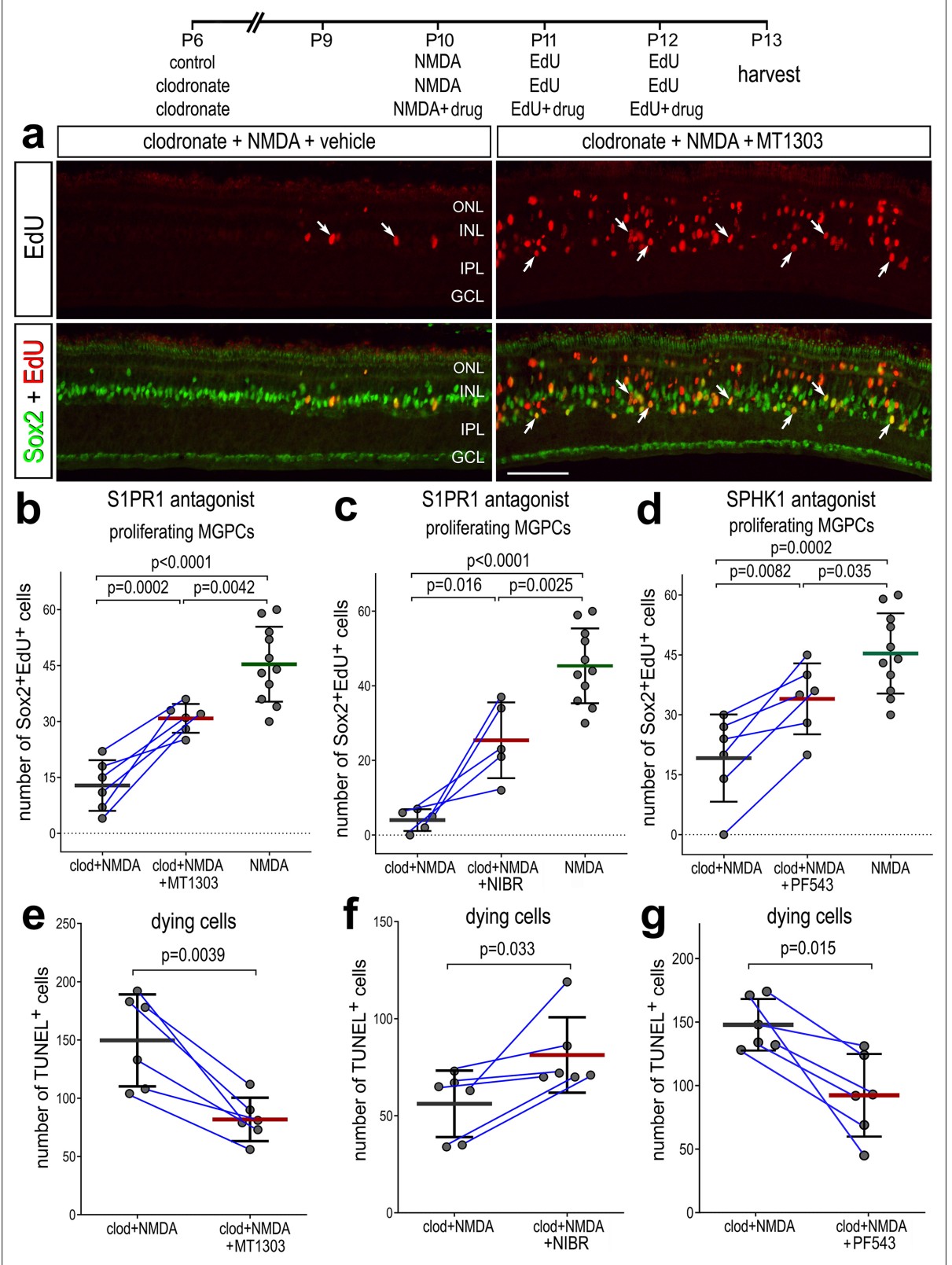

**Figure 8.** Rescue of Müller glia-derived progenitor cell (MGPC) proliferation in damaged retinas missing microglia. Arrows indicate Müller glia (MG), small double arrows indicate EdU-labeled microglia, and hollow arrow-heads indicate proliferating NIRG cells (**a**). The calibration bar represents 50 μm. Histograms (**b–g**) illustrate the mean (bar ± SD), each dot represents one biological replicate and blue lines connect replicates from control and treated retinas from one individual. Significance of difference (p-values) was determined by using a paired *t*-test. Abbreviations: ONL – outer nuclear layer,

*Figure 8 continued on next page*

*Figure 8 continued*

INL – inner nuclear layer, IPL – inner plexiform layer, GCL – ganglion cell layer, NIRG – non-astrocytic inner retinal glial cell, ns – not significant, EdU – 5-ethynyl-2-deoxyuridine.

different S1PR1 inhibitors. Similar to retinas where microglia were present (*Figure 7—source data 1*), there were significantly fewer TUNEL+ nuclei in retinas treated with MT1303 compared to numbers seen in control clodronate/NMDA retinas (*Figure 8e*). However, NIBR0213-treated retinas without microglia contained higher numbers of TUNEL+ nuclei (*Figure 8f*), suggesting that the two S1PR1 inhibitors may have different specificities or toxicities. We found that SPHK1 inhibitor-treated retinas had less cell death compared to clodronate/NMDA retinas (*Figure 8g*). In summary, these findings indicate that S1P signaling is regulated by signals produced by microglia in the retina, and inhibition of S1P signaling is partially rescues the proliferation of MGPCs and reduces cell death in microglia-depleted retinas.

In retinas where microglia have been ablated, levels of *S1PR1* may be transcriptionally maintained by microglia:MG communication in resting conditions. One of the top inferred ligand:receptor interactions identified between resting microglia and MG is TGFB1:CD109 (*El-Hodiri et al., 2023*). Additionally, we have previously shown that TGFβ2 suppresses MGPC proliferation via pSmad3; inhibition of Smad3 by the small molecule inhibitor SIS3 stimulates MGPC proliferation in damaged retinas (*Todd et al., 2017*). Thus, we tested whether TGFβ/Smad3 signaling regulates levels of *S1PR1* in resting MG. We intravitreally delivered SIS3 to retinas at P7 and P8, then harvested retinas at P9. Consistent with earlier findings (*Figure 2*), numerous *S1PR1* FISH puncta were associated with Sox2+ MG nuclei in the middle of the INL in undamaged retinas (*Figure 9a*). SIS3 treatment significantly reduced numbers of *S1PR1* puncta in the INL of undamaged retinas (*Figure 9a, b*). These findings suggest that TGFβ/Smad3 signaling between microglia and MG maintains high levels of *S1PR1*, and the proliferation-inducing effects of Smad3-inhibition may, in part, be mediated by downregulation of S1P signaling in MG.

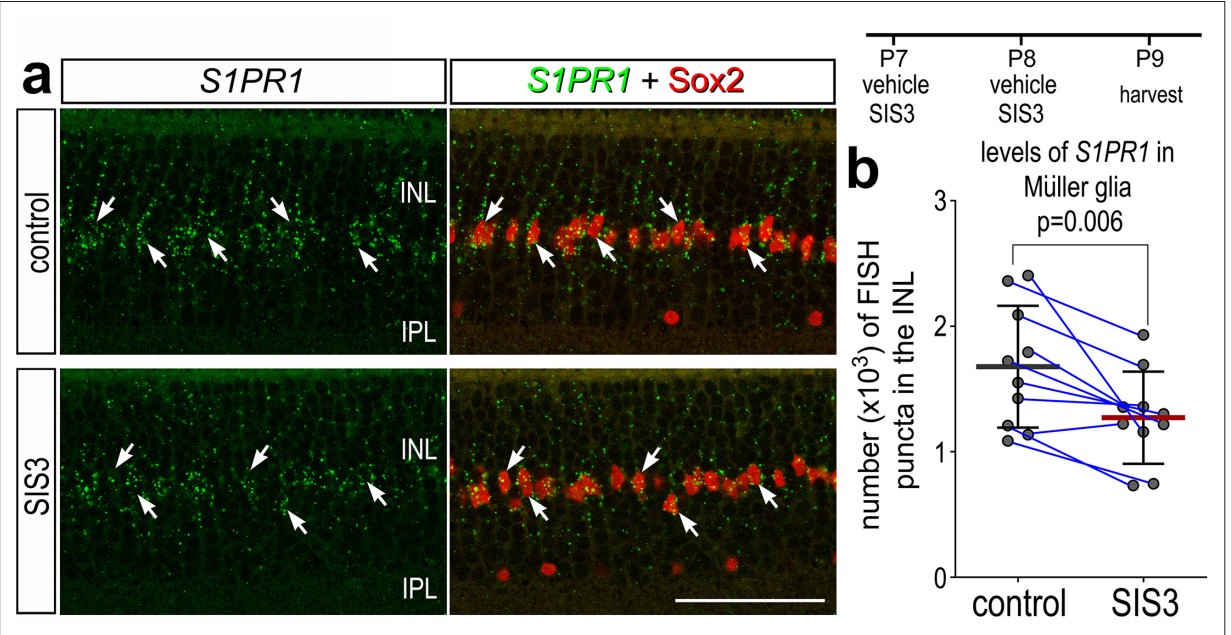

**Figure 9.** Smad3 inhibitor application suppresses S1PR1 transcription. Retinas were obtained from eyes injected with vehicle or SIS3 at P7 and P8, and harvested at P9. Retinal sections were labeled with antibodies to Sox2 (red) and FISH probes to *S1PR1* (green puncta; **a**). Solid arrows indicate MG nuclei associated with numerous FISH puncta. Calibration bar represents 50 μm. Histograms (**c–j**) illustrate the mean (bar ± SD), each dot represents one biological replicate and blue lines connect replicates from control and treated retinas from one individual. Significance of difference (p-values) was determined by using a paired *t*-test. Abbreviations: ONL – outer nuclear layer, INL – inner nuclear layer, IPL – inner plexiform layer, GCL – ganglion cell layer, MG – Muller glia, ns – not significant.

## Patterns of expression for S1P-related genes in human and zebrafish retinas

Comparative analyses of expression have proven useful in implicating important genes with highly conserved patterns of expression. For example, we have reported that NFIA and NFIB show similar patterns of expression in MG of zebrafish, chicks, mice, and large mammals early after retinal damage (*Clark et al., 2019*; *El-Hodiri et al., 2022*; *Hoang et al., 2020*). However, *Nfib* is upregulated in non-neurogenic mouse MG returning to a resting state after NMDA, whereas *NFIA/B/X* expression is lowered in late neurogenic MGPCs of chicks and zebrafish (*Hoang et al., 2020*). These comparative studies led to the development of *Nfia/b/x;Rbpj* cKO mice in which ~45% of targeted MG transdifferentiate into neurons without retinal damage (*Le et al., 2023*). Accordingly, we sought to compare patterns and levels of expression of S1P-related genes in human and zebrafish retinas.

We interrogated snRNA-seq libraries of human retinas generated by Dr. Rui Chen's group (*Li et al., 2023*). UMAP ordering of cells revealed discrete clusters of all major retinal cell types (*Figure 10a*). A large distinct cluster of MG was distinguished based on high levels of expression of *GLUL* and *RLBP1* (*Figure 10b*). Some of these MG expressed genes associated with an activated phenotype including *TGFB2* and *HBEGF,* and these cells coincided with diminished expression of *GLUL* and *RLBP1* (*Figure 10b, c*). *S1PR1* and *S1PR3* were detected in putative resting and activated MG, respectively, whereas *S1PR2* was not widely expressed (*Figure 10d*). *SGPL1* was detected in rod photoreceptors and cells scattered among the different clusters of inner retinal neurons and MG (*Figure 10e*). *SPHK1* was detected in putative activated MG (*Figure 10e*). Additionally, *S1PR3* and *SPHK1* were detected in human retinal astrocytes (*Figure 10d, e*). In summary, expression patterns for S1P receptors, *SPHK1,* and *SGPL1* in human retinas were similar to those seen in chick retinas.

We next interrogated snRNA-seq libraries of zebrafish retinas recently described and generated by the *Lyu et al., 2023*. UMAP ordering of this large aggregate library revealed distinct clusters of all major retinal cell types (*Figure 10f*). MG, MGPCs, and differentiating progeny formed distinct clusters and 'branches' of UMAP-ordered cells (*Figure 10f–h*), and these clusters contained cells from different times after light damage and NMDA-treatment (not shown). Resting MG formed a distinct cluster of cells that expressed high levels of *glula* and *rlbp1a* (*Figure 10g*). The resting MG were contiguous with a 'branch' of cells that extended in MGPCs that expressed *top2a* and *ascl1a* (*Figure 10h*). The UMAP 'branches' of cells represent a continuum of phenotypes shifting from resting MG to MGPCs to differentiation of progeny stretched across different times after light- and NMDA-damage. The MGPCs were contiguous and extended into branches that included differentiating amacrine cells, bipolar cells, rod photoreceptors, and cod photoreceptors that expressed *neurod1* and *neurod4* (*Figure 10h*). High levels of *s1pr1* were observed in resting MG and were downregulated in MGPCs (*Figure 10i*). In addition, *s1pr1* was observed in a branch of cells projecting from the MGPCs into putative differentiating amacrine cells (*Figure 10i*). By comparison, *s1pr2* was detected in microglia and horizontal cells (*Figure 10i*). *s1pr3a* and *s1pr3b* were not detected at significant levels (not shown). Similarly, *sphk1* was detected in very few cells in the retina (*Figure 10j*). By comparison, *sphk2* expression was detected in bipolar cells and relatively few inner retinal neurons (*Figure 10j*). *sgpl1* was detected in microglia and a few cells scattered among the different clusters of inner retinal neurons and rod photoreceptors (*Figure 10j*); *SGPL1* appeared to be more widely expressed by different cells in chick and human retinas. Although *s1pr1* and *sphk1* expression patterns were similar to those seen in chick and human retinas, zebrafish retinal cells also expressed *s1pr2* and *sphk2. s1pr2* was detected in horizontal cells and microglia (*Figure 10i*) and *sphk2* was detected in bipolar cells (*Figure 10j*). Collectively, these findings indicate that S1PR1 is expressed by resting MG in the retinas of different vertebrate classes, indicating conservation of expression patterns, and possibly functions, of S1P signaling and degradation in the vertebrate retina.

## Discussion

Pro-inflammatory signaling acts differentially across species to influence the regenerative capacity of MG after damage. It is important to understand how pro-inflammatory signaling fits into the complex network of pathways that control MG reprogramming. We provide evidence that S1P signaling is among the network of pro-inflammatory pathways the regulate the responses of MG to retinal damage. Here, we report patterns of expression of S1P-signaling components and the effects of activating

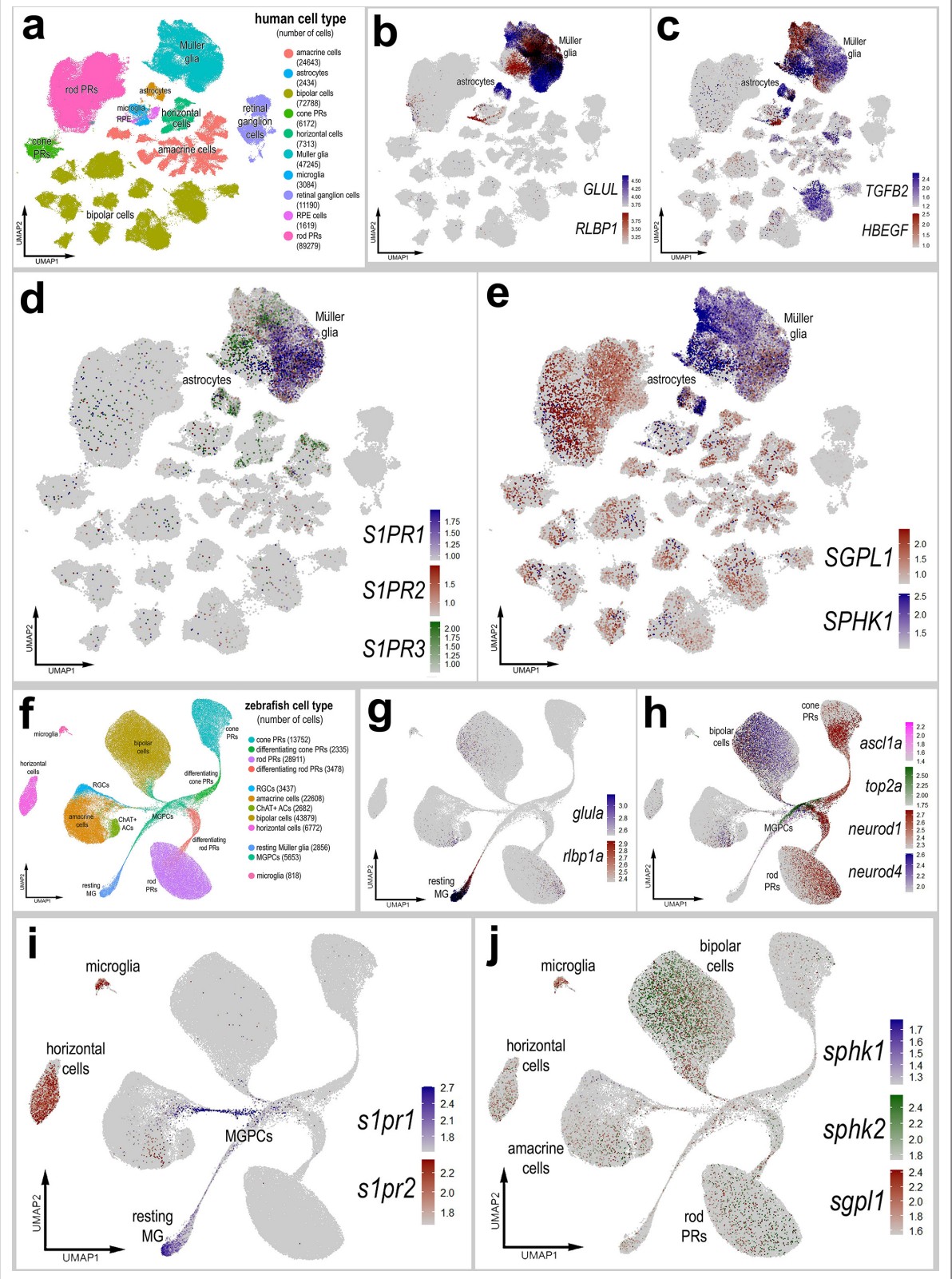

**Figure 10.** Patterns of expression of S1P-related genes in human and zebrafish retinas. scRNA-seq was used to identify patterns of expression of S1P-related factors among retinal cells with the data presented in UMAP plots. Aggregate snRNA-seq libraries were generated for human retinas cells at different ages ranging from 42 to 89 years of age (**a–e**), as described previously (**Li et al., 2023**). Heatmap plots illustrate the patterns and levels of expression for *GLUL* and *RLBP1* (**b**), *TGFB2* and *HBEGF* (**c**), *S1PR1*, *S1PR2*, and *S1PR3* (**d**), and *SGPL1* and *SPHK1* (**e**) across retinal cells. Aggregate

*Figure 10 continued on next page*

*Figure 10 continued*

snRNA-seq libraries were generated for zebrafish retinas cells from eyes treated with vehicle, light damage or NMDA and harvest at 1.5, 2.25, 3, 4, 7, and 14 days after damage (**f–j**), as described previously (**Lyu et al., 2023**). Heatmap plots illustrate the patterns and levels of expression for *glula* and *rlbp1a* (**g**), *ascl1a*, *top2a*, *neurod1*, and *neurod4* (**h**), *s1pr1* and *s1pr3* (**i**), and *sgpl1*, *sphk1*, and *sphk2* (**j**) across retinal cells. Abbreviations: NMDA – *N*-methyl-D-aspartate, UMAP – uniform manifold approximation and projection.

or inhibiting S1P signaling on regeneration in the chick retina. This study highlights the significant role of S1PR1, which is highly expressed in resting MG and non-neurogenic MGPCs, in suppressing damage-dependent MGPC proliferation and neuronal differentiation. Inhibition of S1PR1 promoted MGPC cell-cycle re-entry and enhanced the number of regenerated amacrine-like cells after retinal damage. Inhibition of S1P synthesis produced similar effects to inhibiting S1PR1 to promote MGPC formation, and inhibiting S1P degradation produced similar effects to activating S1PR1 to inhibit MGPC formation.

## Specificity of drugs

We report cell-specific effects of different agonists and antagonists that increase or decrease S1P signaling that is centered on the MG. Since we compare cell-level effects within contralateral eyes wherein one retina is exposed to vehicle and the other is exposed to vehicle plus drug, it seems highly likely that the drugs are responsible for eliciting effects upon the MG. It seems unlikely that the cellular responses that we describe resulted from drugs acting at extra-retinal tissues which second-arily release factors that selectively impact the MG and elicit cellular responses that are inferred by the different agonists/antagonists and consistent with patterns of expression for S1P-related genes. For example, using scRNA-seq and FISH, we find that *S1PR1* and *S1PR3* expression is predominant in MG. Thus, we expect that S1PR1/3 inhibitors directly act on MG and this is consistent with readouts of cell signaling with injections of S1P in undamaged retinas. We show that *SPHK1* and *SGPL1*, which encode the enzymes that synthesize or degrade S1P, are expressed by different retinal cell types, including the MG. The efficacy of the drugs that target SPHK1 and SGPL1 was supported by findings that inhibition of S1P synthesis significantly decreased levels of S1P in normal retinas, whereas inhibition of S1P degradation increased levels of S1P in damaged retinas. Further, inhibition of SPHK1 (to decrease levels S1P) results in decreased levels of ATF3, pS6 (mTor), and pSMAD1/5/9 in MG, consistent with the notion that reduced levels of S1P in the retina impacts signaling at S1PR1/3 that are expressed by MG. Finally, we find similar cellular responses to chemically different agonists or antagonists that target the same protein, and we find opposite cellular responses to agonists and antagonists that target the same protein, which is expected to be complimentary if the drugs are acting specifically at the intended targets.

## Regulation of S1PR1 expression

The identity of factors that downregulate S1PR1 in MG in damaged retinas remains uncertain. Similarly, the identity of the factors that maintain S1PR1 expression in MG in undamaged resting retinas remains uncertain. Previous studies have found S1P production and S1P receptor activation are regulated by several growth factor receptors, including EGFR and VEGFR (**Igarashi et al., 2003**; **Paugh et al., 2008**; **Sukocheva et al., 2006**). In the current study, FGF and insulin treatment were sufficient to downregulate *S1PR1* without retinal damage. Further, we found that TGFβ-dependent Smad3 activity stimulates *S1PR1* expression in resting retinas. These findings support prior in vitro studies which report that TGFβ treatment robustly drives upregulation of S1PR1-3 mRNA and protein (**Lin et al., 2022**; **Yang et al., 2018**). We have previously reported that TGFβ1 is expressed by microglia and TGFβ3 is expressed by neurons in resting retinas, and TGFβ-treatment in damaged retinas suppresses MGPC formation (**El-Hodiri et al., 2023**; **Todd et al., 2017**). Thus, homeostatic TGFβ/Smad2/3 signaling between microglia and MG may maintain *S1PR1* expression to promote quiescence in MG.

Several TFs have been found to influence *S1PR1* expression. KLF2, FOXF1, STAT1, and YAP have been found to directly bind to the S1PR1 promoter to positively regulate transcription (**Cai et al., 2016**; **Carlson et al., 2006**; **Tao et al., 2023**; **Xin et al., 2020**; **Zheng et al., 2023**). Notably, due to the sequestering of NFκB co-activators by KLF2, S1PR1 transcription is linked to the silencing of NFκB-related pro-inflammatory genes (**Jha and Das, 2017**). Conversely, S1PR1 activity has been shown to promote downstream NFκB-mediated gene transcription (**Rostami et al., 2019**). Thus,

signaling through S1PR1 is part of an intricate bidirectional regulatory network that imparts influence through the NFκB pathway. Taken together with present findings, this is consistent with the reported roles of NFκB signaling in rapid microglia-mediated activation of MG and thereafter suppression of MGPC formation in damaged retinas (*Palazzo et al., 2020*).

S1P synthesis in MG in damaged retinas may be a symptom of glial maturation and loss of the ability to reprogram into progenitor-like cells. Unlike MG in peripheral regions of retina, *SPHK1* is not increased in MG and inhibition of SPHK1 does not stimulate the formation of MGPCs in central regions of retina (*Figure 4—figure supplement 4*). Similarly, we have previously reported that Notch-related genes show different patterns of expression in the central and peripheral regions of the retina, and expression levels differentially change at P0, P7, and P21 (*Ghai et al., 2010*). We reported that Notch inhibition reduced cell death and numbers of MGPCs in central regions but not peripheral regions. Because retinal differentiation occurs in a central-to-peripheral gradient and zones of MGPC proliferation become increasingly confined to the periphery in post-hatch birds (*Fischer, 2005*), we hypothesize that maturing MG undergo a shift in maturity or reprogramming competence in a central-to-peripheral gradient. Notch and S1P signaling may be components of this maturation process, but spatial transcriptomic analyses are needed to reveal key differences in signaling pathway gene expression which confer a persistent competence for MG-mediated regeneration in the periphery of chick retinas.

## S1PR1 activation and downstream inflammatory signaling

We and others have reported that inflammatory cytokine signaling is necessary for the initiation of MG cell-cycle re-entry in chicks and zebrafish (*Fischer et al., 2014*; *Silva et al., 2020*), but persistent microglial reactivity and NFκB signaling suppress MG reprogramming (*Palazzo et al., 2020*; *White et al., 2017*). In the current study, we report that S1PR1 activity suppresses MGPC proliferation and neurogenesis without impacting microglia. We report differential activation of Jak/Stat, mTOR, and Smad pathways in MG treated with exogenous S1P or SPHK1 inhibitor. These pathways are activated following NMDA-treatment and are necessary for MGPC formation (*Todd et al., 2016a*; *Todd et al., 2017*; *Zelinka et al., 2016*).

S1PR1 inhibitors and NFκB inhibitors produce similar effects on MGPC proliferation in damaged retinas (*Palazzo et al., 2020*). In the absence of microglia, S1PR1 inhibition promoted damage-dependent MGPC formation but application of an NFκB *activator* was found to increase MGPCs. A comparison of these findings may support the notion that NFκB signaling must be transiently activated to initiate dedifferentiation but must be later suppressed to release the glial phenotype and permit cell-cycle re-entry. It is probable that S1PR1 activation potentiates downstream NFκB signaling to suppress regeneration, but we could not confirm this in the current study; we have not identified antibodies to NFκB pathway components that produce plausible patterns of expression in chick retinas.

Although S1P signaling has been shown to activate the NFκB pathway, we find distinct differences between the impacts of S1P and NFκB signaling on cell signaling in MG, neuronal differentiation of the progeny of MGPCs and neuronal survival in damaged retinas. In the current study we demonstrate that intravitreal injections of S1P activate mTor (pS6) and Jak/Stat3 (pStat3) signaling in MG. Further, inhibition of S1P synthesis decreased ATF3, mTor (pS6) and pSmad1/5/9 levels in activated MG in damaged retinas. By contrast, inhibition of NFκB signaling in damaged chick retinas did not impact the above-mentioned cell signaling pathways (*Palazzo et al., 2020*). However, we cannot exclude the possibility of cross-talk between NFκB and different cell signaling pathways. Further, inhibition of NFκB signaling potently decreases numbers of dying cells and increases numbers of surviving ganglion cells (*Palazzo et al., 2020*). Consistent with these findings, a TNF orthologue, which activates NFκB signaling, exacerbates cell death in damage chick retinas (*Palazzo et al., 2020*). By contrast, five different drugs targeting S1P signaling had no effect on numbers of dying cells and only one S1PR1 inhibitor modestly decreased numbers of dying cells (current study). Although different inhibitors of NFκB-signaling suppressed the proliferation of microglia in damaged retinas (*Palazzo et al., 2020*), all of the S1P-targeting drugs had no effect upon the proliferation of microglia (current study). In addition, inhibition of NFκB does not influence the neurogenic potential of MGPCs (*Palazzo et al., 2020*), whereas inhibition of S1P receptors or inhibition of S1P synthesis increased the differentiation of amacrine-like neurons in damaged retinas (current study). Collectively, our findings indicate that S1P signaling through S1PR1 and S1PR3 in MG has distinct effects upon cell signaling

pathways, neuronal regeneration and cell survival compared to the effects of pro-inflammatory cytokines and NFκB-signaling in the retina.

## S1P and neuronal differentiation

We found that inhibitors to SPHK1, S1PR1, or S1PR1/3 increased numbers of cells that differentiated as amacrine-like cells that expressed HuC/D. However, none the progeny of MGPCs differentiated into cells that expressed calretinin (a subset of amacrine cells) or visinin (photoreceptors). There is currently no evidence that MGPCs in the chick retina can produce progeny that differentiate into photoreceptors. By contrast, we recently reported that inhibition of ID factors enhances neuronal differentiation of calretinin-expressing MGPC-derived cells (*Taylor et al., 2024*). Calretinin may be expressed by a subset of amacrine cells not specified by inhibition of S1P receptors. Alternatively, the progeny of MGPC do not fully differentiate to the point where calretinin is expressed when S1P signaling is diminished. Control over the differentiation of MGPC progeny is regulated by different cell signaling pathways. For example, activation of retinoic acid receptors stimulates neuronal differentiation (*Todd et al., 2018*), whereas signaling through Notch (*Ghai et al., 2010*; *Hayes et al., 2007*), glucocorticoid receptors (*Fischer et al., 2014*), and Jak/Stat3 pathway (*Todd et al., 2016b*) suppress neuronal differentiation. It is possible that activation of S1P receptors acts through Jak/Stat3 or possibly NFκB to suppress the neuronal differentiation of MGPC progeny. Our data indicate that *S1PR1* is downregulated by activated MG and proliferating MGPCs. However, *S1PR3* is upregulated in MGPCs and, thus, it is likely that signaling through S1PR3 mediates the cell-fate specification of MGPC progeny.

## S1P signaling and neuroprotection

The current study provides evidence that inhibition of the S1P pathway supports neuron survival in the damaged chick retinas. In retinas where microglia were absent, we found that SPHK1 inhibitor PF543 and S1PR1 inhibitor MT1303 alleviated retinal cell death. In retinas where microglia were present, only MT1303 significantly reduced numbers of dying cells. As mentioned previously, NFκB-related genes are regulated in a similar pattern to S1P-related genes; inhibiting NFκB signaling supports inner retinal cell and retinal ganglion cell survival (*Palazzo et al., 2020*). In contrast, we recently reported that Inhibitor of DNA-binding 4 (ID4) factor is transiently upregulated in MG after NMDA and may support the survival of inner retinal neurons (*Taylor et al., 2024*). Other factors such as fatty acid binding proteins (FABPs) and Midkine are also rapidly upregulated in activated MG and these factors are neuroprotective (*Carlson et al., 2006*; *Campbell et al., 2022*).

Interestingly, one of the S1PR1 inhibitors, NIBR0213, *increased* cell death in damaged retinas where microglia were absent. This disparity might be explained by differences in inhibitor specificity. MT1303 is a potent functional antagonist which induces receptor internalization and degradation, whereas NIB0213 is slightly less potent and does not drive internalization (*Quancard et al., 2012*; *Shimano et al., 2019*). Further, MT1303 is a prodrug (requiring phosphorylation) and has an elimination half-life of several days, whereas direct-acting NIBR0213 has a half-life of a few hours (*Quancard et al., 2012*; *Shimano et al., 2019*). Studies in mice suggest that sphingosine kinase and receptor inhibition ameliorate the death of retinal ganglion cells following NMDA-treatment (*Basavarajappa et al., 2023*; *Nakamura et al., 2021*). Interestingly, studies have described the neuroprotective effects of fingolimod (S1PR1 modulator) in models of glaucoma in rodents (*Shiwani et al., 2021*). However, we did not find any effect of fingolimod on cell death.

## S1P activity in retinal neurons and microglia

Although *S1PR1* expression was predominantly observed in MG, S1P-signaling components were not restricted to these glial cells. We found that SPHK1, which catalyzes the synthesis of S1P from sphingosine, was expressed by cells scattered across different types of retinal cells. Further, SGPL1, which catalyzes the degradation of S1P, had dynamic patterns of expression following NMDA-treatment in different types of retinal cells, including amacrine cells and microglia. These finding suggest that S1P is secreted by these cells to bind to S1P receptors on MG. S1P:S1PR paracrine signaling has been observed between neural cells and astrocytes and is hypothesized to be necessary for proper astrocyte morphogenesis during synapse formation (*Alam et al., 2023*; *Singh et al., 2022*). Another possibility is that S1P produced by SPHK2. However, SPHK2 is not annotated in the *Gallus gallus* genome,

and we could not identify any genes that aligned significantly with *SPHK2* from other bird species. Further, this gene was not expressed at significant levels in scRNA libraries created from mouse, zebrafish, or pig retinas. A final possibility is that changes in S1P metabolism in retinal neurons are a byproduct of the recycling of other sphingolipid molecules in response to NMDA damage.

### The sphingolipid 'rheostat' in the retina

S1P synthesis is dependent on the availability of ceramides and ceramide-related enzymes like acid ceramidase (ASAH1). Whereas S1P-signaling triggers cell-cycle re-entry, ceramide accumulation drives cellular senescence (*Simón et al., 2019*). Similarly, the protective effects of S1P are countered by ceramides, which activate intrinsic and extrinsic apoptotic pathways (*Simón et al., 2019*). There is significant evidence that ceramide production must be tightly restricted by acid ceramidase, which converts ceramide into sphingosine, for retinal neurons to survive excitotoxic and hypoxic stress (*Lewandowski et al., 2022*; *Nakamura et al., 2021*; *Sugano et al., 2019*; *Yu et al., 2019*). Multiple groups have found that Cer 16:0 and CERS5 are markers for astrocytic inflammation and that astrocytic ceramide production is stimulated by TNF-α (*de Wit et al., 2019*; *Fan et al., 2021*; *Kimura et al., 2000*). In the current study, we observed dynamic regulation of *ASAH1* expression after damage in the chick retina. However, application of a small molecule inhibitor to acid ceramidase had no significant effect on numbers of MGPCs or dying cells (*Figure 4—figure supplement 3*). Many studies have described the effects of ceramides on neuron survival, but additional research is needed to investigate the functional role of ceramides as part of the inflammatory and apoptotic signaling cascades in glial cells.

### Conclusions

In summary, our findings indicate that S1P:S1PR1 signaling plays significant roles in the initiation of MG dedifferentiation, formation of proliferating of MGPCs and specification of progeny toward a neurogenic identity. Our findings suggest that S1PR1 activity maintains quiescence in MG and acts to drive activate toward a resting phenotype. The expression of S1PR1 is regulated by microglia, neuronal damage and TGFβ/Smad3 signaling. Inhibitors to S1P synthesis or S1PR1 signaling promoted MGPC cell-cycle re-entry and enhanced neural differentiation. Interestingly, patterns of expression of S1P-related genes are highly conserved across vertebrates, namely high levels of expression of S1PR1 in the MG in chick, fish and human retinas. Activation of S1PR1 in MG is one of the key pro-inflammatory signaling pathways that control glial responses to neuronal damage, microglial activation, and the ability of MG to reprogram into neurogenic progenitor-like cells.

## Methods

### Animals

The animal use approved in these experiments was in accordance with the guidelines established by the National Institutes of Health and IACUC at The Ohio State University (protocol # 2009A0139-R5). Newly hatched P0 wildtype leghorn chicks (*Gallus gallus domesticus*) were obtained from Meyer Hatchery (Polk, Ohio). Post-hatch chicks were maintained in a regular diurnal cycle of 12 hr light, 12 hr dark (8:00 AM to 8:00 PM). Chicks were housed in stainless-steel brooders at 25°C and received water and Purina chick starter ad libitum.

### Intraocular injections

Chicks were anesthetized with 2.5% isoflurane mixed with oxygen from a non-rebreathing vaporizer. The technical procedures for intraocular injections were performed as previously described (*Fischer et al., 1998*). With all injection paradigms, both pharmacological and vehicle treatments were administered to the right and left eyes, respectively. Compounds were injected in 20 μl sterile saline. Hydrophobic compounds were injected with 20% DMSO in saline, according to solubility data provided by the vendors. Sphingosine and sphingosine-1 phosphate were injected with 0.05 mg/ml bovine serum albumin added as a carrier. 5-Ethynyl-2'-deoxyuridine (EdU) was injected into the vitreous chamber to label proliferating cells. Compounds included in this study are described in *Table 1*. For chicks between P7 and P14, we estimated volumes of 100 μl of liquid vitreous and 800 μl gel vitreous, and an average eye weight of 0.9 g (*Oishi and Murakami, 1985*; *Prashar et al., 2009*). *Table 1* includes the ranges of reported in vivo ED50's (50% effective dose; mg/kg) and the calculated initial maximum

**Table 1.** Pharmacological compounds.

| Drug name | Dose (µg) | Source | Catalog number | ED50 (mg/kg) | Initial max dose (mg/kg equivalent per eye) | Vehicle |
|---|---|---|---|---|---|---|
| NMDA | 73 | Sigma | M3262 | N/A | 81.1 | Saline |
| 5-Ethynyl-2'-deoxyuridine | 2 | Thermo Fisher | A10044 | N/A | 2.2 | Saline |
| Sphingosine (d18:1) | 5 | Avanti | 860490P | N/A | 5.5 | Saline 1% BSA |
| Sphingosine-1 phosphate (d18:1) | 5 | Cayman | 62570 | 0.02 | 5.5 | Saline 1% BSA |
| Amiselimod (MT1303) | 5 | Cayman | 20970 | 0.1–10.0 | 5.5 | 20% DMSO in saline |
| Defensamide (MHP) | 5 | Selleck | S6512 | N/A | 5.5 | 20% DMSO in saline |
| Fingolimod (FTY720) | 10 | Sigma | SML0700 | 0.1–2.3 | 11.1 | 20% DMSO in saline |
| NIBR0213 | 4 | Cayman | 21513 | 0.2–15.0 | 4.4 | 20% DMSO in saline |
| PF-543 (hydrochloride) | 5 | Cayman | 17034 | 1.0–7.5 | 5.5 | 20% DMSO in saline |
| SKI-II | 5 | Sigma | S5696 | 1.0 | 5.5 | 20% DMSO in saline |
| S1PL-in-31 | 6 | Aobious | AOB31664 | 2.0–10.0 | 6.7 | 20% DMSO in saline |
| SEW2871 | 5 | Cayman | 10006440 | 5–20 | 5.5 | 20% DMSO in saline |
| TY 52156 | 5 | Cayman | 19119 | 0.3–10 | 5.5 | 20% DMSO in saline |
| VPC 23019 | 4 | Cayman | 13240 | 0.1–1.0 | 4.4 | 20% DMSO in saline |

dose (mg/kg equivalent per eye). Doses were chosen based on estimates of the initial maximum ocular dose that were within the range of reported ED50's. The S1P-targetting drugs are classified as BDDCS Class II, indicating low solubility but high cell permeability. Thus, it is highly probable that they diffuse across the inner limiting membrane to act within the retina, but it is also likely that their bioavailability is limited, requiring a higher dose, repeated doses and delivery in 20% DMSO. Injection paradigms are included in each figure.

## Preparation of clodronate liposomes

Clodronate liposomes were prepared as described previously (*Fischer et al., 2014*; *Van Rooijen, 1989*). In short, 50 ng cholesterol (Sigma C3045) and 8 mg L-α-phosphatidyl-DL-glycerol sodium salt (Sigma P8318) were dissolved in chloroform and evaporated into a thin film in a round-bottom flask. Then, the liposome film was dissolved in sterile PBS containing 158 mg dichloro-methylene diphosphonate (Sigma D4434). The clodronate was encapsulated by the liposomes via sonication at 42,000 Hz. The clodronate liposomes in PBS were centrifuged at 10,000 RCF, gently resuspended in 150 ml sterile PBS, and injected 20 µl/eye immediately. Our previous studies have shown that this method ablates >99% of microglia/macrophages 2 days after intraocular injection (*El-Hodiri et al., 2023*).

## Fixation, sectioning, and immunocytochemistry

Retinas were fixed, sectioned, and immunolabeled as described previously (*Fischer et al., 1998*). To identify MG, we labeled sections for Sox2 or Sox9. Antibodies to Sox2 and Sox9 are known to label the nuclei or MG in the INL and the nuclei of astrocytes at the vitread surface of the retina (*Ghai et al., 2010*). None of the observed labeling was due to non-specific labeling of secondary antibodies or

**Table 2.** Antibody table – antigen, dilution, host, and sources of antibodies.

| Antibody | Dilution | Host | Clone/catalog number | Source |
|---|---|---|---|---|
| ATF3 | 1:200 | Rabbit | NBP2-85816 | NOVUS |
| Calretinin | 1:1000 | Rabbit | CR7697 | Swant Immunochemicals |
| CD45 | 1:300 | Mouse | HIS-C7 | Cedi Diagnostic |
| cFos | 1:200 | Rabbit | K-25 | Santa Cruz |
| ERK1/2 | 1:600 | Rabbit | 137F5 | Cell Signaling |
| Glutamine synthetase | 1:1000 | Mouse | 610517 | BD Biosciences |
| HuD/C | 1:300 | Mouse | A21271 | Invitrogen |
| Neurofilament | 1:2000 | Mouse | RT97 | DSHB |
| pHisH3 | 1:600 | Rabbit | 06–570 | Millipore |
| pERK1/2 | 1:800 | Rabbit | 4370 | Cell Signaling |
| pS6 | 1:750 | Rabbit | 2215 | Cell Signaling |
| pSmad1/5/9 | 1:250 | Rabbit | D5B10 | Cell Signaling |
| pSTAT3 | 1:300 | Rabbit | 9131 | Cell Signaling |
| Sox2 | 1:1000 | Goat | KOY0418121 | R&D Systems |
| Sox9 | 1:2000 | Rabbit | AB5535 | Millipore |
| Visinin | 1:50 | Mouse | 7G4 | DSHB |

auto-fluorescence because sections labeled with secondary antibodies alone were devoid of fluorescence. Primary antibodies used in this study are described in *Table 2*. Secondary antibodies included donkey-anti-goat-Alexa488/594/647 (Life Technologies A11055; A11058; A21447), donkey-anti-rabbit-Alexa488/594 (Life Technologies A21206; A21207); and goat-anti-mouse-Alexa488/568 (Life Technologies A11001; A-11004) diluted to 1:1000 in PBS plus 0.2% Triton X-100. Nuclear staining was accomplished using DRAQ5 (Thermo 62251).

## Labeling for EdU

For the detection of nuclei that incorporated EdU, immunolabeled sections were fixed in 4% formaldehyde in 0.1 M PBS pH 7.4 for 5 min at room temperature. Samples were washed for 5 min with PBS, permeabilized with 0.5% Triton X-100 in PBS for 1 min at room temperature and washed twice for 5 min in PBS. Sections were incubated for 30 min at room temperature in a buffer consisting of 100 mM Tris, 8 mM $CuSO_4$, and 100 mM ascorbic acid in $dH_2O$. The Alexa Fluor 647 Azide (Thermo Fisher Scientific A10277) was added to the buffer at a 1:500 dilution.

## Terminal deoxynucleotidyl transferase dUTP nick end labeling (TUNEL)

The TUNEL method was used to identify dying cells with fragmented DNA. We used an in-situ Cell Death Detection kit from Roche (Fluorescein, 11684795910) according to the manufacturer's instructions.

## Fluorescent in situ hybridization

Standard procedures were used for FISH, as described previously (*Carlson et al., 2006*). In short, retinas from P9 eyes were fixed for 4 hr RT in 4% paraformaldehyde buffered in 0.1 M dibasic sodium phosphate, washed in PTW (PBS + 0.2% Tween), and incubated in 30% sucrose at 4°C overnight. The retinas were embedded in OCT-compound and cryosectioned at 12 μm. Tissue sections were processed for in situ hybridization with a split-initiator probe pair (Molecular Instruments) according to the manufacturer's protocol for fresh/fixed frozen tissues. For slides in which immunocytochemistry was conducted with FISH, primary antibodies incubated overnight with the hairpin amplification buffer solution, and secondary antibodies incubated for 1 hr the next day. Slides were mounted with glycerol and glass coverslips.

## Measurement of S1P with LC–MS/MS

### Sample preparation

After intraocular injections of saline, NMDA, saline + PF-543, NMDA + PF543, or NMDA + SGPL-in-31, whole retinal tissue samples were collected into a tenfold volume of methanol acidified with 0.1% formic acid. Sphingosine-1-shosphate (d18:1) (S1P) and sphingosine (d17:1) (Avanti Polar Lipids) were used as internal standards. Retinal samples were extracted with 80% methanol at a ratio of 1:20 (wt/vol) in a 2-ml Eppendorf tube. The samples were vortexed for 30 s, sonicated in water bath for 10 min, and followed by centrifugation at 10,000 rcf for 5 min at room temperature. One hundred microliters of supernatant was transferred out to a 2-ml HPLC vial and spiked with sphingosine to a final concentration of 100 ppb. All the samples were analyzed in triplicates.

The matrix blank sample was prepared exactly as samples. Standard addition calibration levels were subsequently prepared by spiking 10, 25, 50, 100, 250, and 500 ppb of S1P into the matrix blank supernatant with final volume of 100 µl. The internal standard of sphingosine was spiked in each calibration levels at 100 ppb. Finally, 5 µl of the standard addition levels were injected and analyzed by LC–MS/MS.

### Quantification

Quantification was carried out on a Vanquish UHPLC coupled to an Orbitrap Exploris 480 mass spectrometer (Thermo Fisher, MA, USA). The analytes were separated on a Accucore C18 2.6 µm 2.1 × 100 mm column using the binary solvents of 0.1% formic acid in water (vol/vol) (solvent A), and 0.1% formic acid in acetonitrile (vol/vol) (solvent B). The gradient was as follows: 0–0.5 min, holding at 10% B; 0.5–6 min, 10–95% B; 6–8 min, holding at 95% B; 8–8.01 min, 95–10% B; 8.01–10 min, holding at 10% B. The flow rate was 0.4 ml/min.

The following mass spectrometer instrument settings were used: ion source = H-ESI; positive ion = 3500 V; sheath gas = 35; aux gas = 7; ion transfer tube temperature = 320°C; vaporizer temperature = 275°C; HCD collision energy = 50%; RF lens = 40%. The S1P and sphingosine were quantified using the transition from 380.2560 to 264.27 $m/z$, and transition from 286.2741 to 268.26 $m/z$, respectively.

### Results

The standard addition curves demonstrated great linearity with $R^2$ above 0.99. Significance of difference was determined using a one-way ANOVA with Šidák correction for multiple comparisons.

### Photography, immunofluorescence measurements, and statistics

Wide-field photomicroscopy was performed using a Leica DM5000B microscope equipped with epifluorescence and Leica DC500 digital camera. Confocal images were obtained using a Leica SP8 imaging system at the Department of Neuroscience Imaging Facility at the Ohio State University. Images were optimized for color, brightness, and contrast, multiple channels over laid and figures constructed using Adobe Photoshop. Cell counts were performed on representative images. To avoid the possibility of region-specific differences within the retina, cell counts were consistently made from the same region of retina for each dataset. Peripheral regions of the retina were designated as 1–3 mm from the circumferential marginal zone, and central regions of the retina were designated as within 3 mm of the posterior pole of the eye in the nasotemporal plane.

The sample size for this study was determined through a power analysis. Animal subjects were randomly assigned to control and treatment groups. Where significance of difference was determined between two treatment groups accounting for inter-individual variability (means of treated-control values) we performed a two-tailed, paired $t$-test. Where significance of difference was determined between two treatment groups, we performed a two-tailed, unpaired $t$-test. Significance of difference between multiple groups was determined using ANOVA followed by Tukey's test. GraphPad Prism 6 was used for statistical analyses and generation of histograms and bar graphs.

### scRNA-seq

We analyzed scRNA-seq libraries that were generated and characterized previously (*Clark et al., 2019*; *Campbell et al., 2022*; *Campbell et al., 2021a*; *El-Hodiri et al., 2022*; *El-Hodiri et al., 2023*; *El-Hodiri et al., 2022*; *Hoang et al., 2020*; *Li et al., 2023*; *Campbell et al., 2021b*). Dissociated cells

were loaded onto the 10X Chromium Cell Controller with Chromium 3'V2, V3, or Next GEM reagents. Using Seurat toolkits (*Powers and Satija, 2015*; *Satija et al., 2015*), UMAP for dimensional reduction plots were generated from nine separate cDNA libraries, including two replicates of control undamaged retinas, and retinas at different times after NMDA-treatment. Seurat was used to construct gene lists for differentially expressed genes, violin/scatter plots, and dot plots. Significance of difference in violin/scatter plots was determined using a Wilcoxon rank sum test with Bonferroni correction. Genes that were used to identify different types of retinal cells included the following: (1) MG: *GLUL*, *VIM*, *SCL1A3*, *RLBP1*, (2) MGPCs: *PCNA*, *CDK1*, *TOP2A*, *ASCL1*, (3) microglia: *C1QA*, *C1QB*, *CCL4*, *CSF1R*, *TMEM22*, (4) ganglion cells: *THY1*, *POU4F2*, *RBPMS2*, *NEFL*, *NEFM*, (5) amacrine cells: *GAD67*, *CALB2*, *TFAP2A*, (6) horizontal cells: *PROX1*, *CALB2*, *NTRK1*, (7) bipolar cells: *VSX1*, *OTX2*, *GRIK1*, *GABRA1*, (8) cone photoreceptors: *CALB1*, *GNAT2*, *GNB3*, *OPN1LW*, and (9) rod photoreceptors: *RHO*, *NR2E3*, *ARR3*. The MG have an over-abundant representation in the scRNA-seq databases. This likely resulted from fortuitous capture-bias and/or tolerance of the MG to the dissociation process.

## Acknowledgements

We thank Dr. Timothy Hla for advice regarding different agonists and antagonists to S1P receptors. We also wish to thank the Mass Spectrometry and Proteomics Core at the Ohio State University Campus Chemical Instrument Center for their services. This work was supported by R01 EY032141-03 (AJF) and R01 EY032141-04 (AJF). Research reported in this publication was also supported by the Ohio State University Comprehensive Cancer Center under NIH Award Number Grant CA016058.

## Additional information

### Funding

| Funder | Grant reference number | Author |
| --- | --- | --- |
| National Eye Institute | R01 EY032141-03 | Andy J Fischer |
| National Eye Institute | R01 EY032141-04 | Andy J Fischer |

The funders had no role in study design, data collection and interpretation, or the decision to submit the work for publication.

### Author contributions

Olivia B Taylor, Conceptualization, Data curation, Formal analysis, Investigation, Visualization, Methodology, Writing – original draft, Writing – review and editing; Nicholas DeGroff, Data curation, Validation; Heithem M El-Hodiri, Writing – review and editing; Chengyu Gao, Data curation, Formal analysis, Writing – original draft; Andy J Fischer, Conceptualization, Formal analysis, Supervision, Visualization, Writing – original draft, Project administration, Writing – review and editing

### Author ORCIDs

Olivia B Taylor https://orcid.org/0009-0002-6827-0552
Heithem M El-Hodiri https://orcid.org/0000-0002-5847-4298
Andy J Fischer https://orcid.org/0000-0001-6123-7405

### Ethics

The animal use approved in these experiments was in accordance with the guidelines established by the National Institutes of Health and IACUC at The Ohio State University (protocol # 2009A0139-R5).

Reviewer #1 (Public review): https://doi.org/10.7554/eLife.102151.4.sa1
Reviewer #2 (Public review): https://doi.org/10.7554/eLife.102151.4.sa2
Author response https://doi.org/10.7554/eLife.102151.4.sa3

# Additional files

## Supplementary files
MDAR checklist

## Data availability
CellRanger output files for Gene-Cell matrices for scRNA-seq data for libraries from saline and NMDA-treated retinas are available at Dryad (https://doi.org/10.5061/dryad.tdz08kq8t), along with the post-hatch normal and treated chick retina databases. scRNA-seq datasets are deposited in GEO (GSE135406, GSE242796) and Gene-Cell matrices for scRNA-seq data for libraries chick retinas treated with saline or NMDA retinas are available through NCBI (GSM7770646, GSM7770647, GSM7770648, GSM7770649). Cell Ranger matrix, features and barcodes files for the snRNA-seq dataset of zebrafish retinas were downloaded through GEO (GSE239410). The Seurat object for the snRNA-seq dataset of human retinas was downloaded through the CELLxGENE collection (https://cellxgene.cziscience.com/collections/4c6eaf5c-6d57-4c76-b1e9-60df8c655f1e).

The following datasets were generated:

| Author(s) | Year | Dataset title | Dataset URL | Database and Identifier |
|---|---|---|---|---|
| Heithem E-H | 2023 | Chick retina, saline control | https://www.ncbi.nlm.nih.gov/geo/query/acc.cgi?acc=GSM7770646 | NCBI Gene Expression Omnibus, GSM7770646 |
| Heithem E-H | 2023 | Chick retina, NMDA treated | https://www.ncbi.nlm.nih.gov/geo/query/acc.cgi?acc=GSM7770647 | NCBI Gene Expression Omnibus, GSM7770647 |
| Heithem E-H | 2023 | Chick retina, clodronate vesicle treated | https://www.ncbi.nlm.nih.gov/geo/query/acc.cgi?acc=GSM7770648 | NCBI Gene Expression Omnibus, GSM7770648 |
| Heithem E-H | 2023 | Chick retina, NMDA + clodronate vesicle treated | https://www.ncbi.nlm.nih.gov/geo/query/acc.cgi?acc=GSM7770649 | NCBI Gene Expression Omnibus, GSM7770649 |
| Taylor O, Degroff N, El-Hodiri H, Gao C, Fischer A | 2025 | Sphingosine-1-phosphate signaling regulates the ability of Müller glia to become neurogenic, proliferating progenitor-like cells | https://doi.org/10.5061/dryad.tdz08kq8t | Dryad Digital Repository, 10.5061/dryad.tdz08kq8t |

The following previously published datasets were used:

| Author(s) | Year | Dataset title | Dataset URL | Database and Identifier |
|---|---|---|---|---|
| Li Q, Dharmat R, Owen L, Shakoor A, Li Y, Kim S, Vitale A, Kim I, Morgan D, Liang S, Wu N, Chen K, DeAngelis MM, Chen R | 2023 | snRNA-seq of human retina - all cells | https://cellxgene.cziscience.com/collections/4c6eaf5c-6d57-4c76-b1e9-60df8c655f1e | Single cell atlas of the human retina, 4c6eaf5c-6d57-4c76-b1e9-60df8c655f1e |
| Wang J, Hoang T, Wang J, Boyd P, Wang F, Hyde DR, Qian J, Blackshaw S | 2020 | Comparative transcriptomic and epigenomic analysis identifies key regulators of injury response and neurogenic competence in retinal glia | https://www.ncbi.nlm.nih.gov/geo/query/acc.cgi?acc=GSE135406 | NCBI Gene Expression Omnibus, GSE135406 |

*Continued on next page*

*Continued*

| Author(s) | Year | Dataset title | Dataset URL | Database and Identifier |
|---|---|---|---|---|
| El-Hodiri HM, Palazzo I, Campbell WA, Fischer AJ | 2023 | Formation of Müller glia-derived progenitor cells in retinas depleted of microglia | https://www.ncbi.nlm.nih.gov/geo/query/acc.cgi?acc=GSE242796 | NCBI Gene Expression Omnibus, GSE242796 |
| Lyu P | 2023 | Common and divergent gene regulatory networks control injury-induced and developmental neurogenesis in zebrafish retina [snRNA-seq] | https://www.ncbi.nlm.nih.gov/geo/query/acc.cgi?acc=GSE239410 | NCBI Gene Expression Omnibus, GSE239410 |

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
