## [Editor Report · eLife Assessment]

This **important** study investigates the signaling pathways regulating retinal regeneration. **Convincing** evidence shows that the sphingosine-1-phosphate (S1P) signaling pathway is inhibited following retinal injury. Small-molecule activators and inhibitors support a model in which S1P signaling must be inhibited to generate Müller glial progenitor cells-a key step in retinal regeneration. The presented results support the major conclusions. However, whether the drug treatments directly or indirectly affect the Müller cells remains unclear.

---

## [Referee Report · Reviewer #1 (Public review)]

Summary:

This study shows that the pro-inflammatory S1P signaling regulates the responses of muller glial cells to damage. The authors describe the expression of S1P signaling components. Using agonist and antagonist of the pathways they also investigate their effect on the de-differentiation and proliferation of Muller glial cells in damaged retina of postnatal chicks. They show that S1PR1 is highly expressed in resting MG and non-neurogenic MGPCs. This receptor suppresses the proliferation and neuronal activity promotes MGPC cell cycle re-entry and enhanced the number of regenerated amacrine-like cells after retinal damage. The formation of MGPCs in damaged retinas is impaired in the absence of microglial cells. This study further shows that ablation of microglial cells from the retina increases the expression of S1P-related genes in MG, whereas inhibition of S1PR1 and SPHK1 partially rescues the formation of MGPCs in damaged retinas depleted of microglia. The studies also show that expression of S1P-related genes is conserved in fish and human retinas.

Strengths:

This is well-conducted study, with convincing images and statistically relevant data

Weaknesses:

In a previous study, the authors have shown that S1P is upstream of NF-κB signaling (Palazzo et al. 2020; 2022, 2023). Although S1P and NF-κB signaling have overlapping effects, the authors here provide evidence for S1P specific effects, adding some new information to the field.

---

## [Referee Report · Reviewer #2 (Public review)]

Summary:

Sphingosine-1-phosphate (S1P) metabolic and signaling genes are expressed highly in retinal Müller glia (MG) cells. This study tested how S1P signaling regulates glial phenotype, dedifferentiation of, reprogramming into proliferating MG-derived progenitor cells (MGPCs), and neuronal differentiation of the progeny of MGPCs using in vivo chick retina. Major techniques used are Sc-RNASeq and immunohistochemistry to determine the gene expression and proliferation of MG cells that co-label with signaling antibodies or mRNA FISH following treating the in vivo eyes with various S1P signaling antagonists, agonists, and signal modulators. The major conclusions drawn are supported by the results presented. However, the methodology they have used to modulate the S1P pathway using various chemical drugs raises questions about the outcomes and whether those are the real effects of S1P receptor modulation or S1P synthesis inhibition.

Strengths:

- Use of elaborated single-cell RNAseq expression data.

- Use of FISH for S1P receptors and kinase as a good quality antibody is not available.

- Use of EdU assay in combination with IHC

- Comparison with human and Zebrafish Sc-RNA data

---

## [Author Response]

The following is the authors’ response to the previous reviews.

**Reviewer #2 (Recommendations for the authors):**
A good number of sentences in the introduction, page two, refer to a figure, 'Fig. 2a', which appears to be the copy-paste effect of these sentences from another location (please see below):"Notably, SPHK2 does not directly contribute to levels of secreted S1P (Thuy et al., 2022), nor is it annotated in the chick genome. S1P can be exported from cells by a transporter (MFSD2A and SPNS2) or converted to sphingosine by a phosphatase (SGPP1) (Fig. 2a). Levels of sphingosine are increased by ASAH1 by conversion of ceramide or decreased by CERS2/5/6 by conversion to ceramide (Fig. 2a). S1P is known to activate G-protein coupled receptors, S1PR1 through S1PR5 (Fig. 2a). S1PRs are known to activate different cell signaling pathways including MAPK and PI3K/mTor, and crosstalk with pro-inflammatory pathways such as NFκB (Fig. 2a) (Hu et al., 2020)."

We have removed references to Fig. 2a, which was from a previous draft of this manuscript.

Please correct the typo in the following sentence (Fid.)"S1PR1 was most prominently expressed by resting MG and MG returning to a resting state, whereas S1PR3 was detected in relatively few scattered cells in clusters of MG, ganglion cells, horizontal cells, bipolar cells, amacrine cells, photoreceptors, oligodendrocytes, microglia and NIRG cells (Fid. 1d).

We have corrected this typo_._